# Towards Automated Knowledge Integration from Human-Interpretable Representations

**Katarzyna Kobalczyk**
University of Cambridge
knk25@cam.ac.uk

**Mihaela van der Schaar**
University of Cambridge
mv472@cam.ac.uk

## Abstract

A significant challenge in machine learning, particularly in noisy and low-data environments, lies in effectively incorporating inductive biases to enhance data efficiency and robustness. Despite the success of informed machine learning methods, designing algorithms with explicit inductive biases remains largely a manual process. In this work, we explore how prior knowledge represented in its native formats, e.g. in natural language, can be integrated into machine learning models in an automated manner. Inspired by the learning to learn principles of meta-learning, we consider the approach of learning to integrate knowledge via conditional meta-learning, a paradigm we refer to as *informed meta-learning*. We introduce and motivate theoretically the principles of informed meta-learning enabling automated and controllable inductive bias selection. To illustrate our claims, we implement an instantiation of informed meta-learning–the Informed Neural Process, and empirically demonstrate the potential benefits and limitations of informed meta-learning in improving data efficiency and generalisation.

## 1 Introduction

**The importance of informed ML and challenges of conventional methods.** The major challenge of machine learning (ML), especially in small data regimes, lies in the selection of an appropriate inductive bias. The hypothesis space of a model must be large enough to encompass a solution to the problem at hand, while also exhibiting a strong preference for solutions that closely align with the ground-truth data generating process (DGP) (Mitchell, 1980; Wilson & Izmailov, 2020). Conventional approaches of inductive bias selection generally rely on external expert knowledge and its manual integration into the learning algorithm, e.g. via feature selection, specialised loss functions, architectures or data augmentation techniques. Despite their successes, such manual knowledge integration methods are limited by the extent to which expert information can be transferred to the learner. Preferences over competing hypothesis can be challenging to formalise and manually integrate into ML methods; with the knowledge integration step often forming the core contribution of many ML papers Goyal & Bengio (2020). Thus, rather than relying on human-engineered ML pipelines, it is natural to seek methods that automatically integrate task-specific knowledge—represented in its native formats, such as natural language—into the learning algorithm. In this paper, we propose the development of such *automated* methods for *controllable* bias selection, meaning that the selected biases are contingent on human-interpretable knowledge representations, facilitating intuitive and steerable specification of inductive biases.

**Meta-learning for automating the selection of inductive biases.** The problem of automatically learning the inductive bias has been previously addressed by introduction of the meta-learning paradigm (Thrun & Pratt, 1998; Baxter, 1997; 2000). In meta-learning, the learner is embedded in an environment of related tasks and the problem of inductive bias learning is seen as the problem of learning a prior over the hypothesis space that enables the learner to generalise to many tasks from this environment. It has been shown that, under certain conditions, a prior learned in this manner is guaranteed to perform well when applied to novel tasks originating from the same environment (Baxter, 2000; Guan & Lu, 2022). However, despite the appeal of meta-learning methods for automated inductive bias selection, they may struggle with heterogeneous task distributions and their performance often drops significantly when faced with out-of-distribution (OOD) tasks Chen et al. (2019); Zhang et al. (2021). Furthermore, priors acquired through meta-learning are largely black-

box in nature, intractably dependent on the choice of meta-training tasks. Once the meta-training phase is concluded, it is not possible to condition the learner on external information specific to the task at hand, and thus failing to satisfy our desideratum of inductive bias steerability.

**Towards automated and controllable inductive bias selection.** Drawing from the ideas of conventional meta-learning, we propose a model of automated and controllable inductive bias specification via knowledge-conditioned meta-learning (Denevi et al., 2020; 2022). Instead of learning a fixed prior $p_\theta(f)$ over the hypothesis space, we consider learning a knowledge-conditioned prior that maps arbitrary representations of expert knowledge, $\mathcal{K}$, to priors, $p_\theta(f|\mathcal{K})$. The learned mapping, $\mathcal{K} \mapsto p_\theta(f|\mathcal{K})$, can be seen as a translator between the human-interpretable knowledge space (e.g. the space of nat-

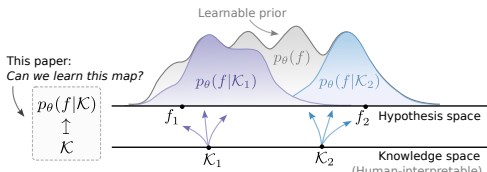

Figure 1: Knowledge representations $\mathcal{K}_i$ condition the heterogeneous learnable prior $p_\theta(f)$. The knowledge-conditioned priors $p_\theta(f|\mathcal{K}_i)$ are more tightly concentrated around the ground truth data-generating functions $f_i$, facilitating stronger inductive biases.

ural language) and the hypothesis space of a model, assigning higher prior probabilities to solutions that are in agreement with the information conveyed in $\mathcal{K}$. We refer to this model in short as **informed meta-learning**. Within this framework, representations of knowledge condition the distribution of tasks and thus inform about task similarity, mitigating the adverse effects of heterogeneous environments of tasks (Fig. 1). As a result, learning with the informed prior, i.e. inferring the posterior $p_\theta(f|\mathcal{D}, \mathcal{K})$ based on an observed dataset, $\mathcal{D}$, can lead to improved data-efficiency, requiring fewer samples of the empirical data to successfully recover the true solution.

The aim of our work is not to present a new method that surpasses existing baselines on a benchmark dataset; rather, we propose a new viewpoint on meta-learning as a means of establishing an interface between human domain knowledge and the hypothesis space of ML models.

**Contributions.** ▶ Section 2: We formalise the interplay between observable data, its underlying generating process and human-interpretable knowledge representations. ▶ Section 3: We introduce a new perspective on meta-learning as an approach enabling automated and controllable inductive bias specification based on the provided representations of knowledge. ▶ Sections 3.2.1-3.2.2: We provide theoretical motivations for this approach and critically discuss the potential opportunities and challenges associated with developing robust informed meta-learners. ▶ Section 4: To illustrate our claims, we implement an instantiation of informed meta-learning—the Informed Neural Process. ▶ Section 5: Through empirical evaluation on both synthetic and real-world datasets, we demonstrate the feasibility of this approach to knowledge integration, as evidenced by improvements in predictive performance and data efficiency. ▶ Section 6: Finally, we discuss future opportunities for informed meta-learning in the context of emergent abilities of foundation models.

## 2 PROBLEM SETTING: THE RELATIONSHIP BETWEEN DATA & KNOWLEDGE

Let $\mathcal{X}$, $\mathcal{Y}$ be the input and output spaces, respectively. A single learning task $\tau = (\mathcal{D}_C, \mathcal{D}_T, \mathcal{K})$ is represented by: a context (aka training) dataset $\mathcal{D}_C = \{(x_i, y_i)\}_{i=1}^n$, a target (aka test) dataset $\mathcal{D}_T = \{(x_i, y_i)\}_{i=1}^m$, and a representation of knowledge about the give learning task $\mathcal{K}$. We assume that context and target datasets are generated according to an underlying function $f : \mathcal{X} \to \mathcal{Y}$, sampled from an unknown stochastic process $f \sim p(f)$, and the observable data is generated according to $\mathcal{D}_C, \mathcal{D}_T \sim p(\mathcal{D}|f)$[1]. The randomness of $p(\mathcal{D}|f)$ captures the unpredictable variabil-

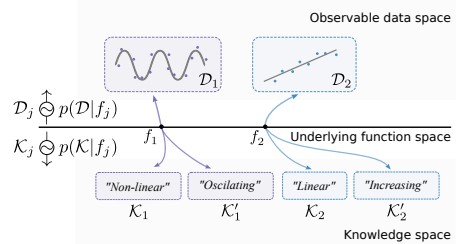

Figure 2: *The generating process of data and knowledge.*

ity of the data generation, given a particular sample function $f$, typically modelled with a Gaussian noise. The knowledge $\mathcal{K}$ is represented in a human-interpretable format, such as natural language, and contains truthful, but likely incomplete information about the underlying $f$. For instance, in the case of 1-dimensional regression, $\mathcal{K}$ could describe the shape of $f$, e.g. $\mathcal{K}$ = *"This function should be non-decreasing"*. We assume that knowledge about a fixed $f$ is generated according to a distribution $p(\mathcal{K}|f)$ which captures the variation of knowledge representations due to their semantic

---

[1]See Appendix A.1.1 for a formal definition of the data and knowledge generating process.

equivalence–a single property of a sample function $f$ can be described by many semantically equivalent expressions in natural language, and due to the incompleteness of information–information expressed in $\mathcal{K}$ is insufficient to unambiguously determine the values of $f$ (e.g. For $\mathcal{K} = $ *"This function is linear"*, we are missing the information about the slope and bias of $f$).

**The goal.** The goal of each learning task is to make predictions on $\mathcal{D}_T$, given $\mathcal{D}_C$ and the additional knowledge $\mathcal{K}$ (see last pane of Fig. 3). With a single task, we would need to rely on a human ML engineer to interpret the information represented in $\mathcal{K}$ and translate it into a model endowed with appropriate inductive biases, penalising (or completely excluding) functions from the hypothesis space of the model that do not conform to the expert knowledge $\mathcal{K}$. This paper considers the following:

*Can we learn a generalisable mapping between human-interpretable knowledge representations and inductive biases of a model? That is, can we learn a map: $\mathcal{K} \mapsto p_\theta(f|\mathcal{K})$, where $p_\theta(f|\mathcal{K})$ assigns approximately zero mass to regions of the hypothesis space that do not conform to the knowledge expressed in $\mathcal{K}$?*

As detailed in the following sections, learning this mapping will necessitate access to additional, related learning tasks $\{\tau_j\}_{j\in\mathcal{J}}$ represented as tuples $\tau_j = (\mathcal{D}_{C,j}\mathcal{D}_{T,j}, \mathcal{K}_j)$, with contexts, targets and knowledge representations generated according to the same process (first two steps in Fig 3) :

$$f_j \sim p(f), \ \mathcal{D}_{C,j}, \mathcal{D}_{T,j} \sim p(\mathcal{D}|f_j), \ \mathcal{K}_j \sim p(\mathcal{K}|f_j).$$

**Knowledge vs. empirical data.** Having in mind the key premise of this paper which is to allow domain experts inject their prior knowledge of a given learning task in an intuitive and flexible manner, we make the following distinctions between knowledge and empirical data: **D1)** The domain of knowledge should be human-interpretable and can be distinct from the domain of empirical data. **D2)** While empirical data is noisy, knowledge should contain only true information. In contrast, additional empirical data or meta-data from potentially alternative modalities, are not a priori known to contain truthful information that is relevant for the given predictive task. **D3)** The way in which information contained in $\mathcal{K}$ is related to the underlying DGP of a task is assumed to be a priori understood by the domain expert. See Appendix A.2.1 for more details and formalization.

## 3 Informed meta-learning

### 3.1 Meta-learning preliminaries

In conventional meta-learning, to enable the learning of the inductive bias, we embed the learner in a distribution of learning tasks. We assume access to a training set of tasks $\{\tau_j\}_{j\in\mathcal{J}}$, represented as tuples of context and target datasets $\tau_j = (\mathcal{D}_{C,j}, \mathcal{D}_{T,j})$. In the probabilistic view, the aim is to learn a posterior predictive map, $\mathcal{A}_\theta : \mathcal{D}_C \mapsto p_\theta(f|\mathcal{D}_C)$. This map can be also interpreted as the learning algorithm. $\mathcal{A}_\theta$ is parametrised by some unknown $\theta \in \Theta$ and we seek $\theta$ that facilitates the best generalisation from context to target data points across all datasets from our training collection. Formally, we find $\theta$ that maximise s:

$$\mathbb{E}_{p(\mathcal{D}_C, \mathcal{D}_T)} \left[ p_\theta(y_T|x_T, \mathcal{D}_C) \right] = \mathbb{E}_{p(\mathcal{D}_C, \mathcal{D}_T)} \int p(y_T|x_T, f) p_\theta(f|\mathcal{D}_C) df, \tag{1}$$

where $p(y_T|x_T, f)$ evaluates the likelihood of target data $x_T, y_T \in \mathcal{D}_T$ for a fixed function $f$ from the hypothesis space. The expectation in (1) is approximated based on the context and target datasets from the training set $\{\tau_j\}_{j\in\mathcal{J}}$. Setting $\mathcal{D}_C = \varnothing$ covers cases where no context data is observed and the learnt distribution $p_\theta(f|\varnothing) = p_\theta(f)$ corresponds to the learnt prior over our hypothesis space , i.e. the learnable inductive bias (grey distribution in Fig. 1). After selecting $\theta$, this prior, however, cannot be adjusted based on potentially available task-specific expert knowledge.

### 3.2 Informed = Knowledge-conditioned meta-learning

To enable the learning of inductive biases that are dependent on external knowledge about the learning task, analogously to the ideas of meta-learning, we embed our learner in a distribution of learning tasks, with each task containing an additional representation of knowledge about the underlying DGP of the data, $\tau_j = (\mathcal{D}_{C,j}, \mathcal{D}_{T,j}, \mathcal{K}_j)$. Analogously to meta-learning, the aim is to learn a posterior predictive map that in this case is knowledge-dependent, $\mathcal{A}_\theta : (\mathcal{K}, \mathcal{D}_C) \mapsto p_\theta(f|\mathcal{D}_C, \mathcal{K})$. By setting $\mathcal{D}_C = \varnothing$, $\mathcal{A}_\theta$ becomes our sought-after map associating arbitrary knowledge representations

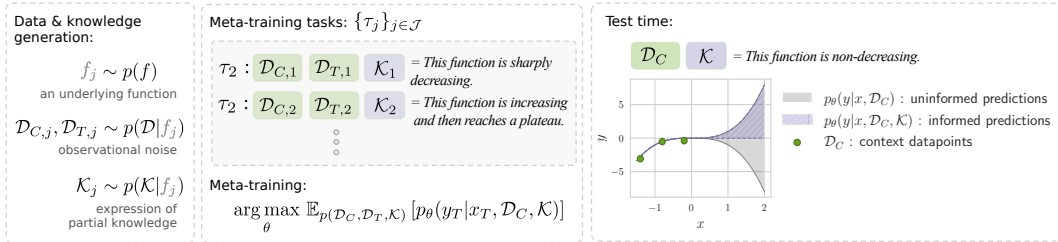

Figure 3: *Informed meta-learning.* Successful knowledge integration via meta-learning ensures that predictions obtained with the informed marginal: $p_\theta(y|x, \mathcal{D}_C, \mathcal{K}_C)$: a) improve upon the uninformed predictions obtained with $p_\theta(y|x, \mathcal{D}_C)$; b) qualitatively reflect our knowledge of the DGP.

$\mathcal{K}$ with an informed prior distribution $p_\theta(f|\mathcal{K})$. This learned prior should reflect the knowledge encoded in $\mathcal{K}$. To learn $\mathcal{A}_\theta$ we should choose $\theta$ that maximises:

$$\mathbb{E}_{p(\mathcal{D}_C, \mathcal{D}_T, \mathcal{K})}\left[p_\theta(y_T|x_T, \mathcal{D}_C, \mathcal{K})\right] = \mathbb{E}_{p(\mathcal{D}_C, \mathcal{D}_T, \mathcal{K})} \int p(y_T|x_T, f)p_\theta(f|\mathcal{D}_C, \mathcal{K})df. \quad (2)$$

Figure 3 summarises the three main steps involved in informed meta-learning. By learning the mapping $\mathcal{K} \mapsto p_\theta(f \mid \mathcal{K})$, informed meta-learning aligns the knowledge space with the hypothesis space of a model (c.f. Fig. 1). At test time, this map can be utilised by a human domain expert to communicate their prior knowledge about a new learning problem in an intuitive way (due to D1-D3). If knowledge is represented in e.g. natural language, this allows for informing the learner about the expected values of predictions in a more flexible way than it would be otherwise possible with additional empirical data. For instance, we may want to convey information about the overall shape of a regression function, rather than just its value at a particular input location. We also note the generality of equations (1) and (2) enabling us to consider multiple incarnations of informed meta-learning based on the existing gradient-based or fully amortised approaches.

**Example 1** (MAML). ▶ **Meta-learning:** The popular gradient-based meta-learning, MAML (Finn et al., 2018), can be recast under the probabilistic formulation of (1) (Grant et al., 2018). In MAML, $\theta$ corresponds to the initialization of the weights of a neural network $f_\theta$ and the map $\mathcal{A}_\theta$ is realised by taking a step of gradient decent with respect to $\mathcal{D}_C$. Given a loss function $\mathcal{L}$, $p_\theta(f|\mathcal{D}_C)$ is a delta function centred at $f_{\theta'}$–the neural network with weights $\theta' = \theta - \nabla\mathcal{L}(f_\theta; \mathcal{D}_C)$, and $p(y_T|x_T, f) \propto \exp(-\mathcal{L}(f; \mathcal{D}_T))$. ▶ **Informed meta-learning** To make the learnable prior of MAML knowledge-dependent, we can modify its original formulation so that the initialization is not common for all tasks, but is conditioned on $\mathcal{K}$. Specifically, we may parametrise the neural network $f_{\omega_\mathcal{K}}$ with weights $\omega_\mathcal{K} = g_\theta(\mathcal{K})$, where $g_\theta$ maps knowledge representations $\mathcal{K}$ to the initialization $\omega_\mathcal{K}$. Therefore, $p_\theta(f|\mathcal{D}_C, \mathcal{K})$ is a delta function centred at $f_{\omega'_\mathcal{K}}$ with $\omega'_\mathcal{K} = \omega_\mathcal{K} - \nabla\mathcal{L}(f_{\omega_\mathcal{K}}; \mathcal{D}_C)$ and $\omega_\mathcal{K} = g_\theta(\mathcal{K})$. During meta-training, we learn the map, $g_\theta : \mathcal{K} \mapsto \omega_\mathcal{K}$. Knowledge representations inform the learner about the similarity of the task at hand with respect to the previously observed learning task; two tasks with the same knowledge are mapped to the same initialization.

**Example 2** (Amortised meta-learners). ▶ **Meta-learning:** Another class of meta-learning methods that can be naturally formulated through (1) are amortised meta-learners for which $\mathcal{A}_\theta$ is reduced to a feed-forward map that embeds $\mathcal{D}_C$ into a fixed vector parametrising $p(y_T|x_T, f)$. Examples include prototypical networks (Snell et al., 2017) or the family of neural processes (Garnelo et al., 2018a;b). ▶ **Informed meta-learning:** Informed extensions of amortised meta-learners can be easily obtained by introducing $\mathcal{K}$ as an additional inputs to the feed-forward map $\mathcal{A}_\theta$.

### 3.2.1 DOES KNOWLEDGE IMPROVE PREDICTIONS?

With informed meta-learning introducing additional inputs in the form of knowledge representations, it is intuitive to think that conditioning the learner on the extra knowledge about the DGP should improve its final predictions on the target data. This is formalised with the following theorem, drawing from the results of Ashman et al. (2024) (see Appendix A.3 for details):

**Theorem 1.** *Suppose that the generating process of datasets and knowledge representations is such that datasets $\mathcal{D}$ and knowledge representations $\mathcal{K}$ are conditionally independent given the underlying $f$. Let $p(y|x, I)$ be the marginal posterior distribution of $y$, given $x$ and additional information $I$, where $I \in \{\mathcal{D}_C, (\mathcal{D}_C, \mathcal{K}), f\}$. Then,*

$$\mathbb{E}_{p(f, \mathcal{D}_C, \mathcal{K})}\left[\text{KL}\left(p(y|x, f)||p(y|x, \mathcal{D}_C, \mathcal{K})\right)\right] \leq \mathbb{E}_{p(f, \mathcal{D}_C)}\left[\text{KL}\left(p(y|x, f)||p(y|x, \mathcal{D}_C)\right)\right], \quad (3)$$

*Proof.* See Appendix A.1.2. □

In the above, the distribution $p(y|x, f)$ is the predictive marginal of $y$ at a fixed location $x$, given the ground-truth $f$ that generated $\mathcal{D}_C$ and that $\mathcal{K}$ describes. In practice, $f$ is unknown and we must infer it based on the available $\mathcal{D}_C$ and optionally $\mathcal{K}$. Theorem 1 suggests that predictions based on $p(y|x, \mathcal{D}_C, \mathcal{K})$ should be closer to the true marginal $p(y|x, f)$ than those based on the uninformed predictive distribution $p(y|, x, \mathcal{D}_C)$. For further theoretical results, including conditions for the inequality (3) to be strict, refer to Appendix A.1.3.

### 3.2.2 REMARKS AND PRACTICAL CONSIDERATIONS

We make several remarks regarding the practical implementation of informed meta-learners.

**Existence of the meta-training set.** The required dataset of tasks consisting of pairs of datasets and their corresponding knowledge representations may seem difficult to obtain, especially at scale. At the same time, the importance of incorporating expert knowledge into ML predictions is greatest where data is scarce. If for the downstream application we are interested in obtaining predictions for a small dataset $\mathcal{D}$, given expert knowledge $\mathcal{K}$, the meta-training set could be obtained from related learning tasks where data is abundant and knowledge easy to obtain. For instance, in medicine, $\mathcal{D}$ could correspond to a classification task of a rare disease and $\mathcal{K}$ describe the typical symptoms as provided by a medical expert. The meta-training set could then consists of classification tasks for more common diseases, where plenty of medical records exist and knowledge about them is easily obtainable, e.g. from medical textbooks. Alternatively, viewing LLMs as databases of knowledge, we can generate synthetic representations of knowledge with LLMs, mimicking the ones expected to be seen at test time. We will employ this technique in Section 5.2.

**Finite sample approximation.** In practice, we model the ground-truth $p(y|x, \mathcal{D}_C, \mathcal{K})$ with $p_\theta(y|x, \mathcal{D}_C, \mathcal{K})$, where the parameters $\theta$ are found by maximising the expected posterior as in equation (2), which is approximated based on a finite meta-training set $\{\tau_j\}_{j \in \mathcal{J}}$. While the inequality in equation (1) holds for the ground-truth marginals, it is no longer guaranteed to hold for $p_\theta$. When the number of training tasks is too small in comparison to the complexity of knowledge representations, empirical approximation of $p(y|x, \mathcal{D}_C, \mathcal{K})$ may fail and the resulting predictions could be worse than those obtained with an uninformed meta-learner trained just on $\{(\mathcal{D}_{C,j}, \mathcal{D}_{T,j})\}_{j \in \mathcal{J}}$. We empirically demonstrate this in Appendix A.8.1 and A.8.2. However, as we will illustrate in section 5, with the size of the meta-training set being sufficiently large, predictions made with the informed posterior $p_\theta(y|x, \mathcal{D}_C, \mathcal{K})$ should result in an improved performance over uninformed predictions, as expected.

**Reduction to meta-learning.** If for a given learning task knowledge is not available, we should expect that the predictions made with an informed meta-learner trained on tasks $\{(\mathcal{K}_j, \mathcal{D}_{C,j}, \mathcal{D}_{T,j})\}_{j \in \mathcal{J}}$ are no-worse than those obtained with an equivalent, uninformed model that has been trained just on $\{(\mathcal{D}_{C,j}, \mathcal{D}_{T,j})\}_{j \in \mathcal{J}}$.

**generalisation & distribution shift.** While generalisation to previously unseen, and semantically novel knowledge representations is highly desirable, it remains a significant challenge, due to the issue of distribution shift. If the underlying knowledge generation process $p(\mathcal{K}|f)$ remains unchanged at test time, then knowledge representations that are semantically novel with respect to those observed during training must necessarily correspond to novel data generating functions $f$ that are OOD in comparison to the training examples.

The next section introduces a practical instantiation of informed meta-learning. This is followed by a set of illustrative experiments aimed at highlighting the above discussion.

## 4 INFORMED NEURAL PROCESSES

To illustrate the concepts discussed in the previous section, we introduce a specific example of an informed meta-learner, Informed Neural Processes (INPs). The proposed class of methods builds on Neural Processes (NPs)—a family of amortised meta-learners first introduced by Garnelo et al. (2018a;b) and later extended with attention-based architectures (Kim et al., 2019; Nguyen & Grover, 2023). NPs reduce the computational cost of learning to a feed-forward operation, eliminating the need for costly gradient-based optimisation. NPs also model a distribution of functions, given the contextual inputs, rather than returning point predictions. This property of NPs is of particular inter-

est to us, as, given the incompleteness of expert knowledge, a single knowledge representation may correspond to multiple underlying functions; a faithful informed meta-learner should adequately represent this form of uncertainty. We remind that the focus of this paper is not to demonstrate state-of-the-art performance on an existing benchmark dataset, but rather to put forward the idea of incorporating expert knowledge into ML predictions via informed meta-learning. Consequently, our implementation remains simple and lightweight, and its details can be found in Appendix A.4.

**The model.** We extend the in-context inputs of conventional NPs with the representations of knowledge pertaining to each learning task and model the predictive posterior as:

$$p_\theta(y \mid x, \mathcal{D}_C, \mathcal{K}) := p_\theta(y \mid x, r_C, k) = \int p_{\theta_d}(y \mid x, z) q_{\theta_e}(z \mid r_C, k) dz, \quad \theta = (\theta_e, \theta_d). \quad (4)$$

In the above, $p_{\theta_d}(y \mid x, z)$ and $q_{\theta_e}(z \mid r_C, k)$ model the conditional distributions $p(y|x, f)$ and $p(f|\mathcal{D}_C, \mathcal{K})$ through a latent variable $z \in \mathbb{R}^d$. The variational distribution $q_{\theta_e}(\cdot|r_C, k)$ depends on both the contextual datapoints $\mathcal{D}_C$ and expert knowledge $\mathcal{K}$ and is modelled with an encoder network $h_{\theta_e}$ consisting of two sub-networks: the data encoder $h_{\theta_{e,D}}$ and knowledge encoder $h_{\theta_{e,K}}$. Fusion of knowledge with data is realised with an aggregation operator $a$, so that $h_{\theta_e}(\mathcal{D}_C, \mathcal{K}) = a(r_C, k)$, with $k = h_{\theta_{e,K}}(\mathcal{K})$ and $r_C = h_{\theta_{e,D}}(\mathcal{D}_C)$. We let $q_{\theta_e}(z|r_C, k) = \mathcal{N}(z; \mu_z, \sigma_z)$ with $(\mu_z, \sigma_z) = a(r_C, k)$ and and we find that choosing $a$ to be a simple sum of the two vectors works well in practice. For regression tasks we model the outputs with a decoder network $g_{\theta_d}$ so that $p_{\theta_d}(y \mid x, z) = \mathcal{N}(y; \mu_x, \sigma_x)$, where $(\mu_x, \sigma_x) = g_{\theta_d}(x, z)$. If either $\mathcal{D}_C$ or $\mathcal{K}$ are not available, we set their respective representations, $r_C$ or $k$, to a zero vector.

**Training.** INPs are trained in an episodic fashion over a distribution of learning tasks $\tau_j = (\mathcal{K}_j, \mathcal{D}_{C,j}, \mathcal{D}_{T,j})$. Omitting the dependence on $j$, we denote by $r_C$ and $r_T$ the representations of context and target data, respectively, and by $k$ the knowledge embedding vector of a single task. Parameters of the model are learned by maximising the expectation of ELBO over all training tasks:

$$\log p_\theta(y_T|x_T, r_C, k) \geq \mathbb{E}_{q_{\theta_e}(z|r_T, k)} \left[\log p_{\theta_d}(y_T \mid x_T, z)\right] - D_{\text{KL}}\left(q_{\theta_e}(z \mid r_T, k) \mid\mid q_{\theta_e}(z \mid r_C, k)\right).$$

During training, we randomly mask knowledge representations by setting $k = 0$. This allows for the possibility of knowledge being missing at test time. Further details on the derivation and estimation of the ELBO loss can be found in Appendix A.5.

## 5 EMPIRICAL STUDY

The experimental section is divided into two parts. First, we illustrate the benefits and challenges associated with informed meta-learning on experiments with synthetic data, where knowledge representations are well-structured and there exists an analytic, closed-form expression linking knowledge with the true DGP. In the second part, we showcase possible applications on real-world data where the underlying DGP is unknown and knowledge may be loosely formatted, particularly, represented in natural language. Full experimental details are described in Appendix A.6.

### 5.1 PART I: ILLUSTRATIVE EXPERIMENTS

Based on the points made in section 3.2.2, we aim to explore the following questions:. **Q1)** Do INPs successfully learn to integrate knowledge into their predictions, and, as a result, improve the data efficiency of the learner? **Q2)** If knowledge is not available at test time, does the performance of the INP model reduce to that of an equivalent NP model? **Q3)** Does the additional knowledge about a learning task help in mitigating the negative affects of task distribution shift? If yes, can the approach of conditional meta-learning enable generalisation to previously unseen and semantically novel knowledge representations? **Q4)** What are the qualitative differences in the impact of knowledge vs. additional empirical data on our predictions?

### 5.1.1 **Q1 & Q2**: KNOWLEDGE AND DATA EFFICIENCY.

**Setup.** For each task, context, and target data points are sampled according to the following process. A function $f$ is sampled from the family of sinusoidal functions with a linear trend and bias, $f(x) = ax + \sin(bx) + c$, for some randomly sampled values of the parameters $a, b, c$. We introduce a Gaussian observational noise, s.t. $y_i = f(x_i) + \epsilon_i$, $\epsilon_i \sim \mathcal{N}(0, 0.2)$. The parameters $a, b, c$ are

randomly sampled according to: $a \sim U[-1,1]$, $b \sim U[0,6]$, $c \sim U[-1,1]$. We let $\mathcal{K}$ encode the value of two, one or none ($\mathcal{K} = \varnothing$) of the parameters $a$, $b$, or $c$. The number of context points $n$ ranges uniformly between 0 and 10; the number of targets is set to $m = 100$. This setup simulates a scenario in which $\mathcal{K}$ contains partial, incomplete information about $f$. By training over a distribution of tasks $\tau$, we expect the model to learn how to put a strong prior on the function's slope, level of oscillations and bias.

**Results.** Fig. 4a shows the estimated log-likelihood on the test tasks against the number of context data points for both the original NP model and the INP. Results for INP are shown with knowledge presented at test time ($\mathcal{K} \neq \varnothing$) and when it is omitted ($\mathcal{K} = \varnothing$). We observe that informing the model significantly improves predictions. As the number of context points decreases, the performance gap between raw and informed predictions increases. Moreover, under $\mathcal{K} = \varnothing$, INPs performs on par with vanilla NPs. Thus, the ability to condition the prior on expert knowledge is not at the cost of reduced performance of purely data-driven predictions.

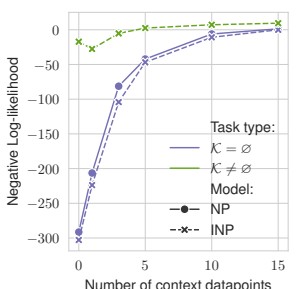

(a) *Avg. log-likelihood vs. number of context points.*

To summarise the impact of knowledge on the predictive performance of INPs we compute the relative $\Delta$AUC score defined as the integral of the "$\Delta$-likelihood against $n$" (von Rueden et al., 2023), with "$\Delta$-likelihood" is defined as: $p(\mathcal{D}_T | \mathcal{D}_C, \mathcal{K})$ - $p(\mathcal{D}_T | \mathcal{D}_C)$. We report relative values with respect to the AUC of the uninformed predictions. Fig. 4b shows the estimated $\Delta$AUC depending on which of the parameters $a$, $b$, or $c$ have their values revealed at test time. Intuitively, exposing more information about $f$ should provide the model with stronger priors, simplifying the learning problem. As expected, when $|\mathcal{K}| = 2$ the performance gains are larger than when $|\mathcal{K}| = 1$.

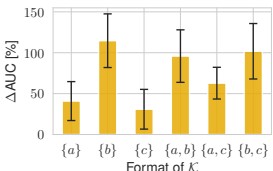

(b) *Relative improvement by knowledge type.*

Figure 4.

### 5.1.2 Q3: DISTRIBUTION SHIFT AND GENERALISATION OF KNOWLEDGE

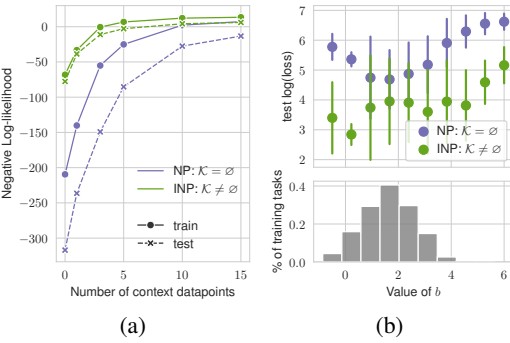

(a)  (b)

Figure 5: a) Average log-likelihood on training vs. testing tasks. b) Top: Log(loss) on testing tasks. Bottom: Frequency of tasks for a given value of $b$ observed during training. Results presented for zero-shot tasks with $\mathcal{D}_C = \varnothing$. Bars represent 1 standard deviation across the tasks within one bin of the $b$ parameter values. Providing knowledge about the parameter $b$ helps in generalisation to OOD tasks. INP generalises to previously unobserved values of $b$.

**Setup.** Performance of meta-learners often drops drastically in the presence of a distribution shift between training and testing tasks Chen et al. (2019). In this experiment, we simulate a distribution shift. Keeping everything else equal as in the previous setup, for the training tasks, we sample $b \sim \mathcal{N}(2,1)$, and for testing tasks we let $b \sim \mathcal{N}(3,1)$. We let $\mathcal{K}$ encode the true value of $b$.

**Results.** Fig. 5a shows how the performance gap between training and testing tasks is significantly reduced upon informing the model about the true value of $b$–the source of the distribution shift between training and testing tasks. From Fig. 5b, we observe that as at test time, as we take the value of the parameter $b$ further out of the range observed during training, the performance of the uninformed model degrades, while the INP maintains its improved performance. Thus, in this instance, INPs successfully generalise to previously unseen knowledge representations.

### 5.1.3 Q4: QUALITATIVE IMPACT OF KNOWLEDGE AND UNCERTAINTY REDUCTION.

The first two columns of Fig. 6 show sample functions from the trained INP from experiment 5.1.1 given a single data point in $\mathcal{D}_C$, and given the knowledge representation (see Fig. A.1 in the appendix for more examples). We find that qualitatively, predictions obtained with the INP correctly reflect the semantic meaning of knowledge representations, with the sampled functions approximately representing the provided values of the parameters $a$, $b$, or $c$. As expected, knowledge

provides information about the global behaviour of sampled functions while individual data points anchor the predictions in the x-y plane.

NPs possess a key feature: the capability to sample from the solution space, instead of providing a single point estimate. This enables us to measure the decrease in model uncertainty when incorporating expert knowledge. Our focus is primarily on measuring epistemic uncertainty—the uncertainty stemming from a lack of knowledge about the true functional relationship, and not the inherent randomness of the data-generating process.

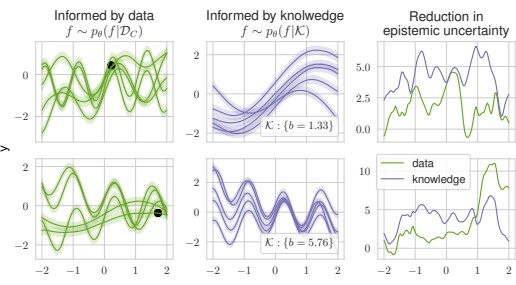

Figure 6: *Sample functions from a trained INP and reduction in epistemic uncertainty.*

To measure the impact of contextual inputs on uncertainty reduction of the INP, we can estimate the predictive uncertainty as the conditional entropy, $\mathbb{H}[p_\theta(y \mid x, I)]$, at a fixed location $x \in \mathcal{X}$ and $I \in \{\mathcal{D}_C, \mathcal{K}, (\mathcal{D}_C, \mathcal{K})\}$. Predictive uncertainty is measured in the observation space and therefore amounts for the uncertainty associated with observational noise. We can, however, decompose it as a sum:

$$\underbrace{\mathbb{I}(y, f \mid x, I)}_{\text{epistemic}} + \underbrace{\mathbb{E}_{f \sim p_\theta(f \mid I)}[\mathbb{H}[p_\theta(y \mid x, f)]]}_{\text{aleatoric}}$$

and approximate the predictive and aleatoric uncertainties with MC samples. The epistemic uncertainty is then obtained as the difference of the two quantities (see appx. A.7 for more details). The last column of Fig. 6 displays the reduction in uncertainty due to knowledge: $\mathbb{I}(y, f \mid x) - \mathbb{I}(y, f \mid x, \mathcal{K})$, and due to data: $\mathbb{I}(y, f \mid x) - \mathbb{I}(y, f \mid x, \mathcal{D}_C)$, computed at all input locations $x \in [-2, 2]$. As observed from the sample functions, we find that knowledge of the value of the parameter $b$, responsible for the oscillations, provides information about global characteristics of $f$, rather than local, in contrast to the single contextual data point in $\mathcal{D}_C$. Future research could explore active-learning strategies (Rainforth et al., 2023) with the task-specific knowledge queried on the basis of the epistemic uncertainty (Kaddour et al., 2020; Astorga et al., 2024; Kobalczyk et al., 2025).

## 5.2 PART II: REAL DATA AND LOOSELY FORMATTED KNOWLEDGE

For illustrative purposes, representations of knowledge in the previous part were highly structured and with a well-defined relationship to the underlying DGP. Naturally, in such scenarios we would resolve to direct knowledge integration methods e.g., a Bayesian regression model (c.f. Appendix A.8.4). However, the advantages of informed meta-learning become evident when: a) the functions to be learned lack a known, closed-form expression; b) knowledge about the learning task is loosely formatted, making manual integration of prior knowledge a significant challenge.

### 5.2.1 INFORMED WEATHER PREDICTIONS

**Setup:** We use the sub-hourly temperature dataset from the U.S. Climate Reference Network, representing values of the air temperature measured at regular 5-minute intervals. For each task, target observations are uniformly sampled from a 24h time range. Context data points are selected by sub-sampling at most 10, chronologically first samples. This setup enables us to assess extrapolation. We perform independent experiments with two formats of knowledge:

- **A:** For each task, knowledge $\mathcal{K}$ is a vector encoding two values: the minimum temperature and the maximum temperature on the day.

- **B:** For each task, knowledge $\mathcal{K}$ is a synthetically generated "weather forecast" presented in natural language. We generate these with GPT-4 OpenAI et al. (2023) prompted to write two sentences mimicking a weather forecast, based on values from the ground truth temperature measurements.

Fig. 7a shows representative examples of the daily temperature paths from test tasks alongside purely data-driven and informed predictions. NPs capture the general trend of the temperature rising during the day, and then falling down towards the night, but unsurprisingly, fail to accurately extrapolate beyond the observed regions. This is due to a high level of heterogeneity present in the collection of meta-training tasks, which is reflected in the high variability of the sampled functions outside of the observed data range. In terms of the informed predictions, we observe that the information contained

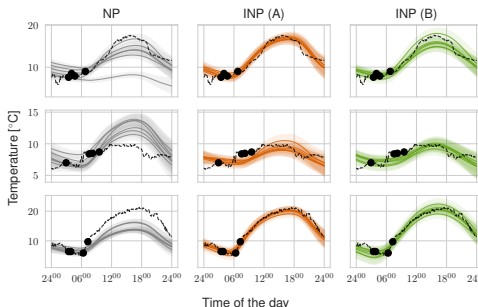

| $n$ | INP **A** | INP **B** |
|---|---|---|
| 0 | 63.5 (1.8) | 60.7 (3.0) |
| 1 | 21.1 (1.4) | 17.3 (1.3) |
| 3 | 14.0 (1.1) | 16.3 (1.3) |
| 5 | 11.1 (1.0) | 13.8 (1.2) |
| 10 | 4.1 (0.5) | 1.1 (0.3) |
| 15 | 0.6 (0.3) | 0.1 (0.2) |
| $\Delta$AUC | 24.6 (1.1) | 22.2 (1.0) |

(a) *Sample predictions.* NP: uninformed, INP (A): informed by knowledge about the minimum and maximum temperature on the day. INP (B): informed by knowledge about the temperature presented in a text format (see appx. A.6)

(b) *Relative performance gap (%) between informed and uninformed predictions.* Bootstrap standard errors in brackets based on 110 testing tasks.

Figure 7.

in $\mathcal{K}$ enables guided extrapolation beyond the observed range of values and reduces the variance of the sampled functions. Table 7b compares the performance gap between informed and uninformed predictions. Notably, knowledge enables sensible, 0-shot predictions with an average improvement in log-likelihood of 57.7% and 48.1% for setups A and B, respectively. We also note that the representation of knowledge, as presented in setup A should, in the theoretically optimal case, impose hard constraints on the maximum and minimum values of the function's range. However, given that INP is only a neural approximation of these constraints, the resulting curves may exceed the specified range as opposed to strictly adhering to it; as it could be possible with a custom-designed model that explicitly incorporates such constraints into its optimisation objective.

**Take-away:** In practical scenarios, predictive functions are difficult to model with closed-form mathematical expressions, making the process of external knowledge integration a challenging task. The benefit of neural, meta- approaches, is their functional flexibility. NPs learn non-trivial "kernels" from the collection of training tasks directly. INPs take this a step further, enabling the incorporation of non-trivially representable information about the underlying function into the model.

### 5.2.2 INFORMED IMAGE CLASSIFICATION

**Setup:** We apply INPs to few-shot classification on the CUB-200-2011 dataset Wah et al. (2011). We use 100 bird categories for training, 50 for validation, and 50 for testing and follow the standard $N$-way, $k$-shot classification setup. We adjust the INP architecture to suit the image classification task, employing CLIP vision and text encoders Fu et al. (2022) (details in appx. A.6.3). We perform independent experiments with three formats of knowledge:

- **A:** Knowledge represents features of a given bird class, e.g. wing span, feather color. Class-level attributes are obtained by averaging the attribute vectors associated with each image from the dataset. Class-level attribute vectors are stacked together to obtain $N \times 312$ tensors.

- **B:** Knowledge represents class-level textual descriptions of the $N$ classes obtained by averaging sentence embedding of individual image captions belonging to the given class. We use human-generated captions as collected in Reed et al. (2016) and embed them with CLIP. Per-class averaged text embeddings are then stacked to form a $N \times 512$ tensor.

- **C:** Here knowledge represents a set of $N$ individual descriptions of each class. We generate these with GPT-4 based on the captions from B (see appx. A.6.3 for examples).

Table 1 displays results for 5-way and 10-way classification. Across all settings, we observe higher classification accuracy when additional knowledge is utilised. This trend holds for 1, 3, 5, and 10-shot tasks, with the performance gap widening as the number of shots decreases. Moreover, the information about characteristic elements of each class contained in $\mathcal{K}$ proves sufficient for relatively good zero-shot prediction performance. While the zero-shot performance for setup C is lower than that of setups A or B, it is nevertheless significantly higher than the accuracy of random guessing.

**Take-away:** INPs align the representations of images and knowledge about class-specific features to construct latent representations that contain the essential multi-modal information. This alignment facilitates robust generalisation to new, previously unseen classes, enabling both zero-shot classification and improved few-shot classification accuracy.

Table 1: *Accuracy (%) on $N$-way, $k$-shot classification tasks for the CUB-200-2011 dataset.* Numbers in brackets represent the bootstrap standard errors of the estimates based on 60 tasks per each setting. Individual tasks are constructed with only previously unseen bird categories.

| | $N = 5$ | | | | $N = 10$ | | | |
|---|---|---|---|---|---|---|---|---|
| $k$ | NP | INP (A) | INP (B) | INP (C) | NP | INP (A) | INP (B) | INP (C) |
| 0 | – | 87.5 (0.7) | 87.4 (0.5) | 50.3 (0.6) | – | 81.1 (0.4) | 78.5 (0.4) | 33.7 (0.3) |
| 1 | 82.2 (0.6) | 88.1 (0.6) | 89.1 (0.5) | 85.1 (0.5) | 73.3 (0.4) | 82.2 (0.4) | 81.9 (0.4) | 77.0 (0.5) |
| 3 | 87.0 (0.5) | 88.4 (0.6) | 89.3 (0.5) | 88.3 (0.5) | 79.8 (0.4) | 82.7 (0.4) | 82.7 (0.4) | 81.8 (0.4) |
| 5 | 88.1 (0.5) | 88.5 (0.6) | 89.6 (0.5) | 88.9 (0.4) | 81.5 (0.4) | 82.8 (0.4) | 82.8 (0.4) | 83.0 (0.4) |
| 10 | 88.5 (0.5) | 88.5 (0.6) | 89.6 (0.5) | 89.0 (0.4) | 82.6 (0.4) | 82.7 (0.4) | 83.0 (0.4) | 84.1 (0.4) |

## 6 RELATED WORK, LIMITATIONS, & IMPACT

**Manual knowledge integration.** Conventional approaches to knowledge integration rely on the ML engineer translating the a priori known property of the underlying function, into an inductive bias of the model. This has been successfully achieved by introducing specialised loss functions (Wu et al., 2018; Karpatne et al., 2017), enforcing informative priors (Mariëlle Zondervan-Zwijnenburg & Schoot, 2017; Constantinou et al., 2016), or designing new model architectures (Cohen & Welling, 2016; Butter et al., 2018). However, such manual approaches are limited by the extent to which expert information can be explicitly encoded in the learning algorithm. Even seemingly straightforward tasks, such as enforcing monotonicity or other shape constraints often require significant engineering effort (Riihimäki & Vehtari, 2010; Link et al., 2022). This restricts representations of expert knowledge to formal mathematical expressions (Karpatne et al., 2017; Qian et al., 2021), knowledge graphs (Choi et al., 2017; Zhang et al., 2019), or logical rules (Yang et al., 2023; Richardson & Domingos, 2006). In contrast, informed meta-learning offers a way to integrate knowledge from any source, including natural language, allowing for a greater flexibility in the type of information that can be transferred to the learner. On the other hand, this approach lacks the guaranteed correctness that conventional methods enjoy. With model-based methods, properties like equivariance or sparsity can be guaranteed by design. Informed meta-learning in its presented form can only approximate the true meaning of knowledge and its performance is contingent on, i.a., the number and coverage of the training tasks, lack of spurious correlations, and complexity of knowledge (see Appendix A.8). Finally, model-based methods do not require access to additional learning tasks that informed meta-learning assumes.

**LLMs to automate knowledge integration.** As LLMs continue to improve, they show potential to act as out-of-the-box, in-context learners, capable of integrating expert knowledge into their predictions. Recent works by Requeima et al. (2024); Jin et al. (2024) explore strategies for eliciting numerical predictions over time series data, given a set of training points and expert information provided as text within the LLM's context window. Such purely LLM-based methods create an interface for human users to incorporate expert insights through language, while also leveraging problem-relevant knowledge embedded in the LLM. These approaches eliminate the need for training an informed meta-learner from scratch. On the other hand, they introduce the risk of incorporating unwanted biases and require transforming numerical datasets into tokens which may limit applicability and performance (Gruver et al., 2023). Future research should explore the mutual benefits of LLM-based learners and informed meta-learning—for instance, by fine-tuning an LLM on the meta-training dataset, or presenting a sample of training tasks within the context window. Such strategies should enable to extend informed learning with LLMs beyond the current focus on 1-dimensional time-series data. LLMs may also serve as a useful tool for synthetic data generation, addressing the limitation of informed meta-learning due to potential difficulties in obtaining the training examples.

An extended discussion of the related work can be found in Appendix A.3.

**Impact.** In this paper, we have proposed a new perspective on automated inductive bias specification based on expert knowledge represented in human-interpretable formats. We note that our work primarily focuses on putting forward the idea of informed meta-learning with the introduced class of INPs serving mainly as an illustration. We hope that this paper inspires future research to explore new architectures improving learning efficiency and generalisation to novel representations of knowledge as well as real-world applications across a range of domains where expert knowledge is present (see A.2.3 for examples). The growing capabilities of LLMs present an exciting opportunity for informed meta-learning, and we look forward to future developments in this area.

## 7 REPRODUCIBILITY AND ACKNOWLEDGEMENTS

**Reproducibility.** Appendix A.4-A.6 contains all the necessary details needed to reproduce the experiments presented in the main body of this paper. We also provide access the anonymous repository containing the source code together with the instructions to reproduce the experiments from sections 5.1 and 5.2.1 (code for the image classification experiments will be made available upon paper acceptance). We also release the synthetically-generated datasets with GPT-4 used in experiments 5.2.1 and 5.2.2. Code and data can be found at: https://github.com/kasia-kobalczyk/informed-meta-learning and at a wider lab repository: https://github.com/vanderschaarlab/informed-meta-learning.

**Acknowledgements.** This work was supported by Azure sponsorship credits granted by Microsoft's AI for Good Research Lab. Katarzyna Kobalczyk is supported by funding from Eedi.

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

# A APPENDIX

Table A.1: Table of Contents

| Appendix section | Contents |
|---|---|
| A.1.1 | Explaining the notation $f \sim p(f)$ and the data and knowledge generating process |
| A.1.2 | Proof of Theorem 1 |
| A.1.3 | Further theoretical results following Theorem 1 |
| A.2.1 | Extended discussion on assumptions D1-D3 regarding data vs. knowledge |
| A.2.2 | A short discussion on the curation of meta-training datasets |
| A.2.3 | Example applications of informed meta-learning |
| A.3 | Extended related work section |
| A.4 | Details of the INP architecture |
| A.5 | Training details and derivation of the ELBO loss |
| A.6 | Details of experiments presented in Section 5 |
| A.7 | Details on uncertainty quantification with INPs |
| A.8 | Additional experiments: A.8.1: the impact of meta-training size on successful knowledge integration; A.8.2 the impact of knowledge complexity on successful knowledge integration; A.8.3: the impact of spurious correlations of knowledge and data on correct "knowledge interpretation"; A.8.4 meta-learned vs. exact knowledge integration. |

## A.1 THEORY

### A.1.1 FORMALISM

Let $(\Omega, \Sigma_\Omega, P)$ be an abstract probability space and $(\mathcal{Y}, \Sigma_\mathcal{Y})$ the measurable space of outputs. We define a stochastic process $F$ as a collection of $\mathcal{Y}$-valued random variables:

$$F := \{F(x) : x \in \mathcal{X}\}.$$

A single realization of $F(\,\cdot\,; \omega) : \mathcal{X} \to \mathcal{Y}$, is a function between the input space $\mathcal{X}$ and the output space $\mathcal{Y}$. We will denote with $p = P \circ F^{-1}$ the law of this stochastic process, so that for a subset $\mathcal{F} \subset \mathcal{X}^\mathcal{Y}$,

$$p(\mathcal{F}) = P(\{\omega \in \Omega : F(\cdot\,; \omega) \in \mathcal{F}\}),$$

Thus, the notation $f \sim p(f)$ corresponds to sampling a function $F(\,\cdot\,; \omega)$ according to the law $p$.

**The data-knowledge generating model.** We assume the following data and knowledge DGP. First, a sample, not directly observable function $f : \mathcal{X} \to \mathcal{Y}$ is sampled according to $f \sim p(f)$. Given $f$, the pair of context and targets datasets is generated according to $\mathcal{D}_C, \mathcal{D}_T \sim p(\mathcal{D}|f)$ and the corresponding knowledge representation is generated according by $\mathcal{K} \sim p(\mathcal{K}|f)$. Here $p(\mathcal{D}|f)$ and $p(\mathcal{K}|f)$ are the data and knowledge generating distributions respectively. Note, this model assumes that knowledge representations and empirical data are conditionally independent given a single realization $f$.

The somewhat unconventional notation of conditioning on functions $f$, through $p(\mathcal{D}|f)$ and $p(\mathcal{K}|f)$ serves to highlight the role of knowledge as expressing information about the true underlying functional relationships between model inputs $x \in \mathcal{X}$ and its outputs $y \in \mathcal{Y}$.

With a slight abuse of notation, we will denote by $p(f, \mathcal{D}_C, \mathcal{K})$ the joint distribution on functions, datasets and knowledge representations. The assumption of conditional independence tells us that it factorise s as $p(f, \mathcal{D}_C, \mathcal{K}) = p(\mathcal{D}_C|f)p(\mathcal{K}|f)p(f)$.

### A.1.2 PROOF OF THEOREM 1

**Theorem 1.** *Suppose that the generating process of datasets and knowledge representations is such that datasets $\mathcal{D}$ and knowledge representations $\mathcal{K}$ are conditionally independent given the underlying $f$. Let $p(y|x, I)$ be the marginal posterior distribution of $y$, given $x$ and additional information $I$, where $I \in \{\mathcal{D}_C, (\mathcal{D}_C, \mathcal{K}), f\}$. Then,*

$$\mathbb{E}_{p(f,\mathcal{D}_C,\mathcal{K})}\left[\mathrm{KL}\left(p(y|x,f)||p(y|x,\mathcal{D}_C,\mathcal{K})\right)\right] \leq \mathbb{E}_{p(f,\mathcal{D}_C)}\left[\mathrm{KL}\left(p(y|x,f)||p(y|x,\mathcal{D}_C)\right)\right], \quad (5)$$

*Proof of Theorem 1 (informal).* The proof is analogous to that presented by Ashman et al. (2024) where $\mathcal{K}$'s take a particular form of an additional sample of data generated according to the same stochastic process as $\mathcal{D}_C$ (see A.3, *Meta-learning with meta-data* for details).

We have that the LHS is equal to:

$$\mathbb{E}_{p(f,\mathcal{D}_C,\mathcal{K})}\left[\mathrm{KL}\left(p(y|x,f)||p(y|x,\mathcal{D}_C,\mathcal{K})\right)\right]$$
$$= \mathbb{E}_{p(f,\mathcal{D}_C,\mathcal{K})}\mathbb{H}\left[p(y|x,f)\right] - \mathbb{E}_{p(f,\mathcal{D}_C,\mathcal{K})}\mathbb{E}_{p(y|x,f)}\left[\log p(y|x,\mathcal{D}_C,\mathcal{K})\right] \quad (6)$$

And that the RHS is equal to:

$$\mathbb{E}_{p(f,\mathcal{D}_C)}\left[\mathrm{KL}\left(p(y|x,f)||p(y|x,\mathcal{D}_C)\right)\right]$$
$$= \mathbb{E}_{p(f,\mathcal{D}_C)}\mathbb{H}\left[p(y|x,f)\right] - \mathbb{E}_{p(f,\mathcal{D}_C)}\mathbb{E}_{p(y|x,f)}\left[\log p(y|x,\mathcal{D}_C)\right] \quad (7)$$

We first consider the second term of (6).

$$-\mathbb{E}_{p(f,\mathcal{D}_C,\mathcal{K})}\mathbb{E}_{p(y|x,f)}\left[\log p(y|x,\mathcal{D}_C,\mathcal{K})\right] =$$
$$= -\mathbb{E}_{p(f,\mathcal{D}_C,\mathcal{K})}\mathbb{E}_{p(y|x,f,\mathcal{D}_C,\mathcal{K})}\left[\log p(y|x,\mathcal{D}_C,\mathcal{K})\right] \qquad \text{(by conditional independence)}$$
$$= -\mathbb{E}_{p(\mathcal{D}_C,\mathcal{K})p(f|\mathcal{D}_C,\mathcal{K})}\mathbb{E}_{p(y|x,f,\mathcal{D}_C,\mathcal{K})}\left[\log p(y|x,\mathcal{D}_C,\mathcal{K})\right]$$
$$= -\mathbb{E}_{p(\mathcal{D}_C,\mathcal{K})}\mathbb{E}_{p(y|\mathcal{D}_C,\mathcal{K})}\left[\log p(y|x,\mathcal{D}_C,\mathcal{K})\right] \qquad \text{(by Foubini)}$$
$$= \mathbb{E}_{p(\mathcal{D}_C,\mathcal{K})}\mathbb{H}\left[p(y|x,\mathcal{D}_C,\mathcal{K})\right]$$
$$\leq \mathbb{E}_{p(\mathcal{D}_C)}\mathbb{H}\left[p(y|x,\mathcal{D}_C)\right] \qquad \text{(conditioning does not increase entropy)}$$

The last expression is equal to the second term of (7):

$$-\mathbb{E}_{p(f,\mathcal{D}_C)}\mathbb{E}_{p(y|x,f)}\left[\log p(y|x,\mathcal{D}_C)\right]$$
$$= -\mathbb{E}_{p(f,\mathcal{D}_C)}\mathbb{E}_{p(y|x,f,\mathcal{D}_C)}\left[\log p(y|x,\mathcal{D}_C)\right] \qquad \text{(by conditional independence)}$$
$$= -\mathbb{E}_{p(\mathcal{D}_C)}\mathbb{E}_{p(f|\mathcal{D}_C)}\mathbb{E}_{p(y|x,f,\mathcal{D}_C)}\left[\log p(y|x,\mathcal{D}_C)\right]$$
$$= -\mathbb{E}_{p(\mathcal{D}_C)}\mathbb{E}_{p(y|x,\mathcal{D}_C)}\left[\log p(y|x,\mathcal{D}_C)\right] \qquad \text{(by Foubini)}$$
$$= \mathbb{E}_{p(\mathcal{D}_C)}\mathbb{H}\left[p(y|x,\mathcal{D}_C)\right]$$

Finally, we notice that the first terms of (6) and (7) are equal:

$$\mathbb{E}_{p(f,\mathcal{D}_C,\mathcal{K})}\mathbb{H}\left[p(y|x,f)\right] = \mathbb{E}_{p(f,\mathcal{D}_C)}\mathbb{H}\left[p(y|x,f)\right],$$

as the entropy term $\mathbb{H}\left[p(y|x,f)\right]$ does not depend on $\mathcal{K}$. This concludes the proof. $\qquad \square$

### A.1.3 A FURTHER THEORETICAL DISCUSSION

**Strict inequality in Theorem 1.** By examining the steps in the proof of Theorem 1, we note that the inequality (5) can be made strict under the assumption of strictly positive mutual information between target values $y$ and knowledge $\mathcal{K}$ at the specified location $x \in \mathcal{X}$, formalised as:

$$\mathbb{I}(y;\mathcal{K}|x) := \mathbb{H}\left[p(y|x)\right] - \mathbb{E}_{p(\mathcal{K})}\mathbb{H}\left[p(y|x,\mathcal{K})\right] > 0 \quad (8)$$

Note, we do not require that for any observed context dataset $\mathcal{D}_C$, $\mathbb{H}\left[p(y|x,\mathcal{D}_C)\right] - \mathbb{E}_{p(\mathcal{K})}\mathbb{H}\left[p(y|x,\mathcal{K},\mathcal{D}_C)\right] > 0$. We may imagine special cases in which given a context dataset $\mathcal{D}_C$, any additional knowledge about the underlying function would be redundant. For instance, in the noiseless setup of section 5.1.1, with $f(x) = ax + \sin(bx) + c$, observing 3 distinct data points fully determines the underlying values of the parameters $a$, $b$, and $c$; thus, any extra knowledge $\mathcal{K}$ would be redundant in this case.

**Proposition 1.** *Under the assumption of (8),*

$$\mathbb{E}_{p(\mathcal{D}_C, \mathcal{K})} \mathbb{H}\left[p(y|x, \mathcal{D}_C, \mathcal{K})\right] < \mathbb{E}_{p(\mathcal{D}_C)} \mathbb{H}\left[p(y|x, \mathcal{D}_C)\right] \qquad (9)$$

*Proof.* Note, we can decompose $\mathbb{E}_{p(\mathcal{D}_C, \mathcal{K})} \mathbb{H}\left[p(y|x, \mathcal{D}_C, \mathcal{K})\right]$ as a sum of two terms:

$$\mathbb{E}_{p(\mathcal{D}_C, \mathcal{K})} \mathbb{H}\left[p(y|x, \mathcal{D}_C, \mathcal{K})\right] =$$
$$= \mathbb{E}_{p(\mathcal{D}_C, \mathcal{K})} \left[\mathbb{1}\{\mathcal{D}_C = \varnothing\}\mathbb{H}\left[p(y|x, \mathcal{D}_C, \mathcal{K})\right] + \mathbb{1}\{\mathcal{D}_C \neq \varnothing\}\mathbb{H}\left[p(y|x, \mathcal{D}_C, \mathcal{K})\right]\right]$$

For the first term, we have:

$$\mathbb{E}_{p(\mathcal{D}_C, \mathcal{K})} \left[\mathbb{1}\{\mathcal{D}_C = \varnothing\}\mathbb{H}\left[p(y|x, \mathcal{D}_C, \mathcal{K})\right]\right]$$
$$= \mathbb{E}_{p(\mathcal{D}_C, \mathcal{K})} \left[\mathbb{1}\{\mathcal{D}_C = \varnothing\}\mathbb{H}\left[p(y|x, \mathcal{K})\right]\right] \qquad (p(y|x, \varnothing, \mathcal{K}) = p(y|x, \mathcal{K}))$$
$$= \mathbb{E}_{p(f)}\mathbb{E}_{p(\mathcal{D}_C|f)}\mathbb{E}_{p(\mathcal{K}|f)} \left[\mathbb{1}\{\mathcal{D}_C = \varnothing\}\mathbb{H}\left[p(y|x, \mathcal{K})\right]\right] \quad \text{(by conditional independence)}$$
$$= \mathbb{E}_{p(f)}\mathbb{P}(\mathcal{D}_C = \varnothing|f)\mathbb{E}_{p(\mathcal{K}|f)}\mathbb{H}\left[p(y|x, \mathcal{K})\right]$$
$$= \mathbb{P}(\mathcal{D}_C = \varnothing)\mathbb{E}_{p(\mathcal{K})}\mathbb{H}\left[p(y|x, \mathcal{K})\right] \qquad (*)$$
$$< \mathbb{P}(\mathcal{D}_C = \varnothing)\mathbb{H}\left[p(y|x)\right] \qquad \text{(by the assumption (8))}$$
$$= \mathbb{E}_{p(\mathcal{D}_C)} \left[\mathbb{1}\{\mathcal{D}_C = \varnothing\}\mathbb{H}\left[p(y|x)\right]\right]$$

The transition in $(*)$ follows since the data generating process is such that the event $\mathbb{1}\{\mathcal{D}_C = \varnothing\}$ is independent of the particular choice of the data generating function $f$ (e.g., in our experiments, the number of available data points for conditioning is sampled uniformly from a range of values regardless of the underlying function).

For the second term, since conditioning does not increase entropy, for any fixed dataset $\mathcal{D}_C$ we have:

$$\mathbb{E}_{p(\mathcal{K}|\mathcal{D}_C)}\mathbb{H}\left[p(y|x, \mathcal{D}_C, \mathcal{K})\right] \leq \mathbb{H}\left[p(y|x, \mathcal{D}_C)\right].$$

Multiplying both sides by $\mathbb{1}\{\mathcal{D}_C \neq \varnothing\}$ and taking the expectation with respect to $p(\mathcal{D}_C)$ yields:

$$\mathbb{E}_{p(\mathcal{D}_C, \mathcal{K})} \left[\mathbb{1}\{\mathcal{D}_C \neq \varnothing\}\mathbb{H}\left[p(y|x, \mathcal{D}_C, \mathcal{K})\right]\right] \leq \mathbb{E}_{p(\mathcal{D}_C)} \left[\mathbb{1}\{\mathcal{D}_C \neq \varnothing\}\mathbb{H}\left[p(y|x, \mathcal{D}_C)\right]\right].$$

Combining the two inequalities gives us:

$$\mathbb{E}_{p(\mathcal{D}_C, \mathcal{K})} \mathbb{H}\left[p(y|x, \mathcal{D}_C, \mathcal{K})\right] =$$
$$= \mathbb{E}_{p(\mathcal{D}_C, \mathcal{K})} \left[\mathbb{1}\{\mathcal{D}_C = \varnothing\}\mathbb{H}\left[p(y|x, \mathcal{D}_C, \mathcal{K})\right]\right] + \mathbb{E}_{p(\mathcal{D}_C, \mathcal{K})} \left[\mathbb{1}\{\mathcal{D}_C \neq \varnothing\}\mathbb{H}\left[p(y|x, \mathcal{D}_C, \mathcal{K})\right]\right]$$
$$< \mathbb{E}_{p(\mathcal{D}_C)} \left[\mathbb{1}\{\mathcal{D}_C = \varnothing\}\mathbb{H}\left[p(y|x)\right]\right] + \mathbb{E}_{p(\mathcal{D}_C)} \left[\mathbb{1}\{\mathcal{D}_C \neq \varnothing\}\mathbb{H}\left[p(y|x, \mathcal{D}_C)\right]\right]$$
$$= \mathbb{E}_{p(\mathcal{D}_C)} \mathbb{H}\left[p(y|x, \mathcal{D}_C)\right]$$

$$\square$$

**Corollary 1.** *Suppose that the generating process of datasets and knowledge representations is such that datasets $\mathcal{D}$ and knowledge representations $\mathcal{K}$ are conditionally independent given the underlying $f$. Let $p(y|x, I)$ be the marginal posterior distribution of $y$, given $x$ and additional information $I$, where $I \in \{\mathcal{D}_C, (\mathcal{D}_C, \mathcal{K}), f\}$. Suppose that in addition, assumption (8) holds. Then,*

$$\mathbb{E}_{p(f, \mathcal{D}_C, \mathcal{K})} \left[\text{KL}\left(p(y|x, f)||p(y|x, \mathcal{D}_C, \mathcal{K})\right)\right] < \mathbb{E}_{p(f, \mathcal{D}_C)} \left[\text{KL}\left(p(y|x, f)||p(y|x, \mathcal{D}_C)\right)\right], \qquad (10)$$

*Proof.* It suffices to replace in the proof of Theorem 1 the inequality $\mathbb{E}_{p(\mathcal{K}|\mathcal{D}_C)}\mathbb{H}\left[p(y|x, \mathcal{D}_C, \mathcal{K})\right] \leq \mathbb{H}\left[p(y|x, \mathcal{D}_C)\right]$ with the strict inequality from Proposition 1. $\square$

**When can we expect assumption (8) to hold true?** Given the above results, it is natural to consider when can we expect $\mathbb{I}(y; \mathcal{K}|x) > 0$ to hold true. It may be in fact the case that $\mathcal{K}$ provides localised information, e.g. $\mathcal{K} =$*"The values for $x$'s smaller than 0 should not exceed ..."*. For this kind of knowledge, if a given $x$ does not fall into the region that $\mathcal{K}$ informs about (in the example, it is any $x \geq 0$), and without enforcing any further restrictions on the underlying process $p(f)$, (8) is may not hold. However, instead of considering the impact of $\mathcal{K}$'s at a specified location $x$, it is reasonable to assume that a weaker conditions holds, i.e. that the conditional mutual information between the targets $y$ and knowledge $\mathcal{K}$ is positive in expectation across the domain $\mathcal{X}$:

$$\mathbb{E}_{p(x)} \left[\mathbb{I}(y; \mathcal{K}|x)\right] > 0,$$

in which case we obtain that:

$$\mathbb{E}_{p(x,f,\mathcal{D}_C,\mathcal{K})}\left[\mathrm{KL}\left(p(y|x,f)||p(y|x,\mathcal{D}_C,\mathcal{K})\right)\right] < \mathbb{E}_{p(x,f,\mathcal{D}_C)}\left[\mathrm{KL}\left(p(y|x,f)||p(y|x,\mathcal{D}_C)\right)\right]. \quad (11)$$

We may also want to further break down the assumption of (8 with respect to the information gain of $\mathcal{K}$ on the underlying $y$ through $f$ rather than directly. Due to the conditional independence of observed values and knowledge given $f$, we have the following:

$$
\begin{aligned}
\mathbb{I}(y;\mathcal{K}|x) &= \mathbb{I}(\mathcal{K};y|x) \\
&= \mathbb{H}[p(\mathcal{K})] - \mathbb{E}_{p(y|x)}\mathbb{H}[p(\mathcal{K}|x,y)] \\
&= \mathbb{H}[p(\mathcal{K})] - \mathbb{E}_{p(f)}\mathbb{H}\left[p(\mathcal{K}|f)\right] + \mathbb{E}_{p(f)}\mathbb{H}\left[p(\mathcal{K}|f)\right] - \mathbb{E}_{p(y|x)}\mathbb{H}[p(\mathcal{K}|x,y)] \\
&= \mathbb{H}[p(\mathcal{K})] - \mathbb{E}_{p(f)}\mathbb{H}\left[p(\mathcal{K}|f)\right] + \mathbb{E}_{p(f)}\mathbb{H}\left[p(\mathcal{K}|f,x,y)\right] - \mathbb{E}_{p(y|x)}\mathbb{H}[p(\mathcal{K}|x,y)] \quad (\dagger) \\
&= \underbrace{\mathbb{H}[p(\mathcal{K})] - \mathbb{E}_{p(f)}\mathbb{H}\left[p(\mathcal{K}|f)\right]}_{=\mathbb{I}(\mathcal{K};f)} - \mathbb{E}_{p(y|x)}\left[\underbrace{\mathbb{H}[p(\mathcal{K}|x,y)] - \mathbb{E}_{p(f)}\mathbb{H}\left[p(\mathcal{K}|f,x,y)\right]}_{=\mathbb{I}(\mathcal{K};f|x,y)}\right] \\
&= \mathbb{I}(f;\mathcal{K}) - \mathbb{E}_{p(y|x)}\left[\mathbb{I}(f;\mathcal{K}|x,y)\right],
\end{aligned}
$$

where we have used the conditional independence assumption in ($\dagger$). Thus, $\mathbb{I}(y;\mathcal{K}|x) > 0$ if and only if:

$$\mathbb{I}(f;\mathcal{K}) - \mathbb{E}_{p(y|x)}\left[\mathbb{I}(f;\mathcal{K}|x,y)\right] > 0, \quad (12)$$

where the above terms can be expressed as: $\mathbb{I}(f;\mathcal{K}) = \mathbb{H}[p(f)] - \mathbb{E}_{p(\mathcal{K})}\mathbb{H}[p(f|\mathcal{K})]$ and $\mathbb{I}(f;\mathcal{K}|x,y) = \mathbb{H}[p(f|x,y)] - \mathbb{E}_{p(\mathcal{K})}\mathbb{H}[p(f|\mathcal{K},x,y)]$ . Therefore, for (12) to be true we require:

1. **Non-zero dependency between $f$ and $\mathcal{K}$.** We must necessarily have that $\mathbb{I}(f;\mathcal{K}) > 0$, which should trivially be true by the definition of knowledge as information about the ground truth data generating process.

2. **Sufficient informativeness of $(x,y)$ about $f$.** Note, $\mathbb{E}_{p(y|x)}\left[\mathbb{I}(f;\mathcal{K}|x,y)\right]$ is always smaller or equal to $\mathbb{I}(f;\mathcal{K})$. The condition of $\mathbb{E}_{p(y|x)}\left[\mathbb{I}(f;\mathcal{K}|x,y)\right] < \mathbb{I}(f;\mathcal{K}|x)$, implies that the observation noise of $y$'s should be low. High noise leads to higher $\mathbb{E}_{p(y|x)}\left[\mathbb{I}(f;\mathcal{K}|x,y)\right]$ bringing it closer to $\mathbb{I}(f;\mathcal{K})$ and thus causing $\mathbb{I}(y;\mathcal{K}|x)$ to go to zero.

**More information leads to stronger priors.** In the experiments, we have observed what is quite intuitive, namely, that providing the model with more information in knowledge representations results in stronger priors and thus improved predictions. To formalise this claim we can assume two competing knowledge-generating distributions: $p(\mathcal{K}|f)$ and $\tilde{p}(\mathcal{K}|f)$, with their corresponding joints: $p(f,\mathcal{D}_C,\mathcal{K}) = p(f)p(\mathcal{D}_C|f)p(\mathcal{K}|f)$ and $\tilde{p}(f,\mathcal{D}_C,\mathcal{K}) = p(f)p(\mathcal{D}_C|f)\tilde{p}(\mathcal{K}|f)$. In this case we have that :

$$\tilde{p}(y|x) = \int\int \tilde{p}(y,f,\mathcal{K}|x)d\mathcal{K}df = \int\int p(f)p(y|x,f)\tilde{p}(\mathcal{K}|f)d\mathcal{K}df = \int p(f)p(y|x,d)df = p(y|x).$$

Then, saying that $p(\mathcal{K}|f)$ is (locally) more informative than $\tilde{p}(\mathcal{K}|f)$ can be formalised as:

$$
\begin{aligned}
\mathbb{H}\left[p(y|x)\right] - \mathbb{E}_{p(\mathcal{K},\mathcal{D}_C)}\mathbb{H}\left[p(y|x,\mathcal{D}_C,\mathcal{K})\right] &> \mathbb{H}\left[p(y|x)\right] - \mathbb{E}_{\tilde{p}(\mathcal{K},\mathcal{D}_C)}\mathbb{H}\left[\tilde{p}(y|x,\mathcal{D}_C,\mathcal{K})\right] \Leftrightarrow \\
\mathbb{E}_{\tilde{p}(\mathcal{K},\mathcal{D}_C)}\mathbb{H}\left[\tilde{p}(y|x,\mathcal{D}_C,\mathcal{K})\right] &> \mathbb{E}_{p(\mathcal{K},\mathcal{D}_C)}\mathbb{H}\left[p(y|x,\mathcal{D}_C,\mathcal{K})\right]. \quad (13)
\end{aligned}
$$

Let,

$$\epsilon(x) := \mathbb{E}_{p(f)p(\mathcal{D}_C|f)p(\mathcal{K}|f)}\left[\mathrm{KL}\left(p(y|x,f)||p(y|x,\mathcal{D}_C,\mathcal{K})\right)\right]$$

and

$$\tilde{\epsilon}(x) := \mathbb{E}_{p(f)p(\mathcal{D}_C|f)\tilde{p}(\mathcal{K}|f)}\left[\mathrm{KL}\left(p(y|x,f)||\tilde{p}(y|x,\mathcal{D}_C,\mathcal{K})\right)\right].$$

It is then straightforward to show that under the assumption of (13), $\epsilon(x) < \tilde{\epsilon}(x)$, i.e. the posterior predictive is closer to the ground-truth marginal $p(y|x,f)$ for knowledge generated according to the more informative distribution.

## A.2 Extended Remarks, Limitations & Future Work

### A.2.1 Knowledge vs. data

We remind the reader that the primary objective of this paper is to consider a learning method that allows domain experts to inject their prior knowledge of a given learning task in an intuitive and flexible manner. In this regard, we elaborate on what distinguishes expert knowledge from empirical (meta-) data.

In line with the goal of intuitive and flexible knowledge integration, D1) posits that the domain of knowledge should be human-interpretable and, as such, distinct from the domain of empirical data. Specifically, we argue that in many real-world applications, knowledge can be most naturally expressed in natural language, which allows for the articulation of task properties that may be difficult to encode using strict mathematical equations, or additional data samples.

D2) asserts that, unlike empirical data, knowledge should consist only of true information. The assumption of truthfulness in knowledge is a recurring theme throughout this paper and warrants further clarification. Referring to our data and knowledge generation process, we assume the following: first, an underlying function $f \sim p(f)$ is sampled. This function $f$ generates both the observable dataset $\mathcal{D}_C \sim p(\mathcal{D}|f)$ and the target dataset $\mathcal{D}_T \sim p(\mathcal{D}|f)$, with $p(\mathcal{D}|f)$ introducing the observational noise. An expert, with privileged insight into the data-generating process, provides a description of the expected properties of $f$ with $\mathcal{K} \sim p(\mathcal{K}|f)$. Variability in $p(\mathcal{K}|f)$ arises due to two factors: 1) For a single function $f$, semantically equivalent representations may be used to express the same properties–this is particularly relevant when $\mathcal{K}$ is represented in natural language; 2) Knowledge about a specific function $f$ is typically partial–only a subset of the properties of the underlying $f$ may be reasonably expected to be known for a given task. The assumption of truthfulness also implies that, up to semantic equivalence, the process $p(\mathcal{K}|f)$ should not be subject to distributional shifts, unlike $p(\mathcal{D}|f)$ or $p(f)$ itself.

Lastly, D3) states that the relationship between the information contained in $\mathcal{K}$ and the underlying DGP of a task is assumed to be understood a priori by the domain expert. This implies that the process $p(\mathcal{K}|f)$ is, in itself, human-interpretable.

Assumptions D1) to D3) are in contrast with the use of additional data in hierarchical meta-learning setups considered in prior works, where contextual data $\mathcal{D}_C$ is supplemented by additional meta-data about the task. For instance, (Iwata & Kumagai, 2022) condition the meta-learner on the feature names of the dataset $\mathcal{D}_C$. However, feature names alone do not convey meaningful, human-interpretable information about the learning task that informs about the underlying relationship between predictors and the target variable. While this relationship may be inferred during meta-training, it would remain unknown to the human expert, violating D3). The relationships between feature names and the underlying values of the variables is also likely to be subject to distribution shifts, violating D2). As a result, even though conditioning on such additional data may lead to improved predictive performance, it would not enable the intuitive steerability of inductive biases during deployment.

### A.2.2 Collection of the meta-training dataset.

In practice, a typical setup may begin with one or a few contextual datasets, each paired with expert knowledge about the underlying functional relationships expressed in $\mathcal{K}$. This raises a natural question: how can additional learning tasks required for informed meta-learning be obtained?

First, we acknowledge that the requirement of the additional training tasks is may be a limitation of informed meta-learning in comparison to model-based strategies. In particular, the number of training tasks required for successful knowledge integration via informed meta-learning grows with the complexity of the knowledge representations, as illustrated in the synthetic experiment in Appendix A.8.2. For the informed meta-learner to understand the underlying meaning of knowledge, it must be exposed to a variety of datasets corresponding to semantically distinct representations of knowledge. Without semantic variation, learning a generalisable mapping between the knowledge space and the hypothesis space of a model becomes impossible.

Given that interviewing a human domain expert to express knowledge about a large number of related learning tasks may be impractical, we propose that LLMs, viewed as repositories of human

knowledge, can be utilised for smart synthetic data and knowledge generation. While the exact methods for generating such data safely and reliably are beyond the scope of this paper, we suggest this as an avenue for future work. Nonetheless, synthetic data generation or augmentation is a viable and promising strategy, as suggested by experiments 5.2.1 and 5.2.2.

### A.2.3 EXAMPLE APPLICATIONS OF INFORMED META-LEARNING

The aim of the experiments presented in section 5.1.3 is not only to validate the empirical effectiveness of our approach but also to illustrate its potential in practical, knowledge-rich domains where natural language insights may improve low-data learning. We can draw parallels between these experiments and other domains. To inspire future directions of applied research we have provided additional example applications of informed meta-learning in Table A.2.

Table A.2: *Example applications of informed meta-learning.*

| Domain | Task type | Data | Knowledge |
|---|---|---|---|
| Finance | Economic / stock market forecasting | GDP over time, or daily stock prices | Descriptions of trends e.g. *"The stock prices for APPL are expected to continue the upward trend"*. |
| Healthcare | Disease classification | Patient health measurements | Expected relationships between symptoms and diseases e.g., *"High blood pressure is often linked to cardiovascular issues."* |
| Medical Imaging | Disease detection in X-rays | Images of lungs | Relationships between abnormalities and diseases e.g., *"Presence of shadows in upper lung areas may suggest tuberculosis."* |
| Energy | Demand forecasting | Hourly electricity consumption data | Expectations based on events or trends e.g., *"Energy demand is expected to spike in winter months due to heating needs."* |
| Logistics | Delivery time estimation | Delivery routes, traffic patterns | Expected effects of traffic and weather e.g., *"Deliveries are typically delayed during peak hours in urban areas."* |
| Retail | Demand prediction | Sales data, time series | Expected shopping behaviours e.g., *"Sales tend to increase during holiday seasons, especially for electronic goods."* |

### A.3 EXTENDED RELATED WORK

The main body of this paper discusses prior work that is related with respect to achieving the goal of knowledge integration. Here we discuss related work from the perspective of the proposed method of informed meta-learning and the concrete instantiation of INPs.

**Conditional generative models.** The goal of deep generative models (DGMs) is to learn a neural approximation of the distribution of the data $p(x)$ over a space $\mathcal{X}$, most commonly the space of images. Popular DGMs include, VAEs Kingma & Welling (2014), GANs Goodfellow et al. (2014), and diffusion models Ho et al. (2020); Song & Ermon (2019). Their conditional versions, e.g. CVAEs Sohn et al. (2015), CGANs Mirza & Osindero (2014), and conditional diffusion models Ho & Salimans (2022); Ramesh et al. (2022) model the conditional distribution $p(x \mid c)$, where $c$ is an additional conditioning variable, e.g. a class label or a text sequence. A similar analogy can be drawn between NPs and INPs which, as meta-learners, bring the idea of (conditional) generative modelling to the space of predictive functions $f : \mathcal{X} \to \mathcal{Y}$. The goal of NPs is to model the prior distribution over functions $p(f)$; as well as the posterior predictive distribution $p(f \mid \mathcal{D}_C)$. Similarly to CVAEs and CGANs, INPs introduce an additional conditioning variable–the expert knowledge, and model the conditional distribution $p(f \mid \mathcal{K})$, guiding the prior over the space of functions, such that the informed predictions, dictated by $p(f \mid \mathcal{D}_C, \mathcal{K})$ are concentrated around the region of functions agreeing with both the observed dataset $\mathcal{D}_C$ and the expert knowledge $\mathcal{K}$.

**Multimodal deep learning.** Mulitmodal deep learning refers to deep learning methods that can process and relate information from multiple modalities simultaneously, such as image, audio, and text. Our framework assumes that the datasets $\mathcal{D}$ and knowledge representations $\mathcal{K}$ may belong to two different data modalities (e.g. $\mathcal{D}$ contains input-output pairs for 1-D regression and $\mathcal{K}$ contains a natural language description of the expected shape of the regression curve). This places informed meta-learning in the area of multimodal methods. What makes it distinct is that standard multimodal strategies (e.g. Ngiam et al. (2011); Srivastava & Salakhutdinov (2012); Ding & Tao (2015); Shi et al. (2021b;a)) consider finding a predictive function $f$, where $\mathcal{X}$ is a multimodal input space $\mathcal{X} = \mathcal{X}_1 \times \ldots \times \mathcal{X}_M$, with each $\mathcal{X}_j, j \in [M]$ corresponding to a different data modality. In informed meta-learning, the learned functions $f$ are typically unimodal, but the learning algorithm to fit each function is conditioned on the knowledge representation $\mathcal{K}$, belonging to a different modality.

**Zero and Few-shot learning.** As presented in the experimental section, informed meta-learning enables sensible zero-shot predictions guided by expert knowledge. For instance, in multi-class image classification, $\mathcal{K}$ may contain a list of characteristic attributes of each class or class-wide descriptions in natural language. Seemingly similar ideas of utilising side information about each class for zero-shot learning have been explored in works of Al-Halah et al. (2016); Elhoseiny et al. (2017); Paz-Argaman et al. (2020). In contrast to these methods, informed meta-learning does not focus on zero-shot learning only, but on the process of integrating external knowledge (e.g. knowledge about what are the characteristic features of each class) with observed few-shot or zero-shot ($\mathcal{D}_C = \varnothing$) data sample. In the image classification domain, the idea of combining sample images with zero-shot attribute information was considered by Tsai & Salakhutdinov (2017) in application to one-shot learning, and extended by Schönfeld et al. (2019) to few-shot learning. None of these works, however, consider the meta-learning setup of $N$-way, $k$-shot classification and require that the class attribute information is always present at training and test time, as it implicitly defines class labels. In our setup, the role of class information contained in $\mathcal{K}$ lies in enhancing model performance by emphasising which visual features are most distinctive for a given class, enabling zero-shot classification as a by-product. Contrary to Schönfeld et al. (2019), the additional information contained in $\mathcal{K}$ is not necessary for few-shot predictions on previously unseen classes.

**Meta-learning with meta-data.** Several studies considered applying meta-learning to hierarchical datasets that include empirical data alongside corresponding metadata, such as feature names Iwata & Kumagai (2022), task-specific parameters Kaddour et al. (2020), or labels Tseng et al. (2022). In these works, metadata is used in conjunction with empirical data as additional inputs defining task similarities. While these works share similarities in terms of the data structure and the resulting method, their motivations and goals are distinct. For instance, Kaddour et al. (2020) extend active learning principles to meta-learning, allowing the algorithm to select which task to learn next based on the task metadata. Huang et al. (2022); Belbute-Peres et al. (2021); Kovacs et al. (2022), apply meta-learning to solve differential equations given a specific parametrisation. Another stream of work utilise s additional data to improve performance and generalisation of meta-learners: Iwata & Kumagai (2022) leverage feature descriptions, and Tseng et al. (2022) use multi-class agricultural labels. Denevi et al. (2020; 2022) study theoretically the conditions under which conditional meta-learning is advantageous to non-conditional meta-learning in the specific case of gradient-based meta-learning and with hypothesis space restricted to linear functions of the input. A recent work by Ashman et al. (2024) conditions the learner on an additional set of datasets, which are known a priori to have been generated by the same stochastic process as the context and target data. This approach aligns with informed meta-learning–it leverages the known relationship between the knowledge representation, e.g. a set of extra datasets, and the underlying DGP of $\mathcal{D}_C$. Our work generalises to arbitrary representations of knowledge, moving beyond structured numerical forms, with a particular focus on human-interpretable representations, such as natural language.
In contrast to the existing works, the motivation of this paper is not on designing a new meta-learning method suitable for hierarchically structured learning tasks. Instead, we view meta-learning as means to enable human experts inject their prior knowledge about a given learning problem into the learning algorithm, in an intuitive and more flexible way than it is possible with conventional, model-based methods. Further discussion on the differences between informed meta-learning and meta-learning with hierarchically structured datasets are discussed in A.2.1.

## A.4   INP Model Architecture

The architecture of INPs consists of the following key components:

- A **data encoder**, $h_{\theta_{e,D}} : (\mathcal{X} \times \mathcal{Y})^* \to \mathbb{R}^d$ that takes in pairs $(x_i, y_i)$ and produces an order-invariant representation $r = \sum_i h_{\theta_{e,D}}(x_i, y_i)$.
- A **knowledge encoder**, $h_{\theta_{e,K}}$, a map from the knowledge representation space to the latent space $\mathbb{R}^d$ that takes in the knowledge inputs $\mathcal{K}$ and extracts a latent knowledge vector $k = h_{\theta_{e,K}}(\mathcal{K})$.
- An **aggregator**, $a$, that combines the data representation, $r$, and the latent knowledge representation, $k$, into one representation that parametrises the latent distribution $q_{\theta_e}$. We take $q_{\theta_e}(z \mid r, k) := \mathcal{N}(z; \mu_z, \sigma_z)$, where $(\mu_z, \sigma_z) = a(r, k)$.
- A **conditional decoder**, $g_{\theta_d}$, that takes in samples of the global latent variable $z \sim q_{\theta_e}(z \mid r, k)$ and the new target location $x$ to output the predictions parametrised by $p_{\theta_d}(y \mid x, z) := \mathcal{N}(y; \mu_x, \sigma_x)$, with $(\mu_x, \sigma_x) = g_{\theta_d}(x, z)$.

In all experiments any MLP is implemented with the GELU non-linearity Hendrycks & Gimpel (2016). Number and dimensions of hidden layers are reported individually for each experiment in section $A.6$. We experiment with different forms of aggregation, $a$:

1. **sum & MLP**: $a(r, k) = \mathrm{MLP}(r + k)$,
2. **concat & MLP**: $a(r, k) = \mathrm{MLP}([r||k])$,
3. **MLP & FiLM**: $a(r, k) = \mathrm{FiLM}(k)\,[\mathrm{MLP}(r)]$. We use the idea of modulation parameters introduced by Perez et al. (2018). Here $a$ is an MLP whose parameters are modulated with a modulated with the outputs of $h_{\theta_{e,K}}$.

We find that in most cases, the first, least complex option performs the best.

## A.5 INP TRAINING

INPs are trained in an episodic fashion over a distribution of learning tasks consisting of context and target datasets, and associated knowledge representations. Denoting by $r_C$ and $r_T$ the context and target data representations and by $k$ the knowledge embedding vector of a single task, we derive the evidence lower bound via:

$$p_\theta(y_T \mid x_T, r_C, k) = \int p_{\theta_d}(y_T | x_T, z) q_{\theta_e}(z \mid r_C, k) dz \tag{14}$$

$$= \int p_{\theta_d}(y_T \mid x_T, z) \frac{q_{\theta_e}(z \mid r_C, k)}{q_{\theta_e}(z \mid r_T, k)} q_{\theta_e}(z \mid r_T, k) dz \tag{15}$$

$$= \mathbb{E}_{q_{\theta_e}(z|r_T,k)} \left[ p_{\theta_d}(y_T \mid x_T, z) \frac{q_{\theta_e}(z \mid r_C, k)}{q_{\theta_e}(z \mid r_T, k)} \right] \tag{16}$$

And therefore, by Jensen we obtain:

$$\log p_\theta(y_T \mid x_T, r_C, k) \geq \mathbb{E}_{q_{\theta_e}(z|r_T,k)} \left[\log p_{\theta_d}(y_T \mid x_T, z)\right] - D_{\mathrm{KL}}\left(q_{\theta_e}(z \mid r_T, k) \,||\, q_{\theta_e}(z \mid r_C, k)\right)$$

The parameters of the model are learned by maximising the above ELBO for randomly sampled batches of tasks. During training, we use one sample of $q_{\theta_e}(z \mid r_T, k)$ to form a MC estimate of the ELBO. For evaluation, we use 32 samples. Additionally, during training, we randomly mask knowledge by setting $k = 0$, the frequency of masking is a hyperparameter of the model.

## A.6 EXPERIMENTAL DETAILS

Throughout all experiments we use the Adam optimise r Kingma & Ba (2015). During training, we use validation-based early stopping. All experiments were run on a machine with an AMD Epyc Milan 7713 CPU, 120GB RAM, and using a single NVIDIA A6000 Ada Generation GPU accelerator with 48GB VRAM.

### A.6.1 1-D SINUSOIDAL REGRESSION (SECTION 5.1)

For each task, context and target data points are sampled according to the following process. A function $f$ is sampled from the family of sinusoidal functions with trend and bias, $f(x) = ax + \sin(bx) + c$. We also introduce a Gaussian observational noise, s.t. $y_i = f(x_i) + \epsilon_i, \epsilon_i \sim \mathcal{N}(0, 0.2)$. The parameters $a, b, c$ are randomly sampled according to: $a \sim U[-1, 1], b \sim U[0, 6], c \sim U[-1, 1]$. For each task, the context and target points are uniformly sampled from the range $[-2, 2]$. The

Table A.3: *Average negative log-likelihood for different knowledge types.* Results averaged across 500 testing tasks with standard errors of the mean provided in the brackets.

| $\mathcal{K}$
$n$ | $\varnothing$ | $\{a\}$ | $\{b\}$ | $\{c\}$ | $\{a,b\}$ | $\{b,c\}$ | $\{a,c\}$ |
|---|---|---|---|---|---|---|---|
| 0 | 292.7 (225.0) | 196.2 (173.4) | 80.9 (122.8) | 222.1 (230.6) | 43.1 (70.0) | 49.3 (85.8) | 106.8 (96.0) |
| 1 | 218.1 (218.3) | 165.9 (170.9) | 47.8 (94.7) | 192.6 (203.1) | 30.3 (57.1) | 40.6 (78.4) | 145.1 (149.7) |
| 3 | 97.4 (139.1) | 75.3 (113.0) | 10.8 (43.7) | 88.7 (146.5) | 3.1 (26.4) | 8.9 (38.6) | 64.8 (94.5) |
| 5 | 47.0 (110.3) | 34.6 (82.3) | -1.6 (27.4) | 45.8 (104.6) | -3.4 (21.4) | -1.5 (26.5) | 31.7 (77.9) |
| 10 | 8.5 (52.2) | 7.2 (49.6) | -9.1 (12.8) | 7.9 (54.8) | -9.3 (11.8) | -9.1 (12.5) | 4.5 (40.6) |
| 15 | 2.2 (33.6) | -0.0 (30.0) | -9.0 (15.4) | 1.5 (33.3) | -9.9 (11.9) | -9.3 (14.4) | -0.3 (28.5) |

number of context points $n$ ranges uniformly between 0 and 10; the number of targets, $m = 100$. We let $\mathcal{K}$ to encode the value of two, one, or none ($\mathcal{K} = \varnothing$) of the parameters $a$, $b$, or $c$.

The data encoder, $h_{\theta_{e,D}}$, is implemented as a 3-layer MLP. The knowledge encoder, $h_{\theta_{e,K}}$, is implemented with the DeepSet architecture Zaheer et al. (2017), made of two 2-layer MLPs. Each element of the set is represented by a one-hot encoding of the parameter type with its value appended at the end. The decoder is a 4-layer MLP. We set the hidden dimension, $d = 128$ and use the sum & MLP method for the aggregator, $a$. We use a learning rate of 1e-3 and set the batch size to 64. During training, knowledge is masked at rate 0.3.

In section 5.1 we use this setup to demonstrate and discuss the impact of expert knowledge on enhanced data-efficiency, reduction in uncertainty, and robustness to distribution shifts. Fig. A.1 shows sample predictions under 0, 1, or 3 observed data points and different formats of knowledge $\mathcal{K}$. Table A.3 provides a detailed summary of the negative log-likelihood by knowledge type.

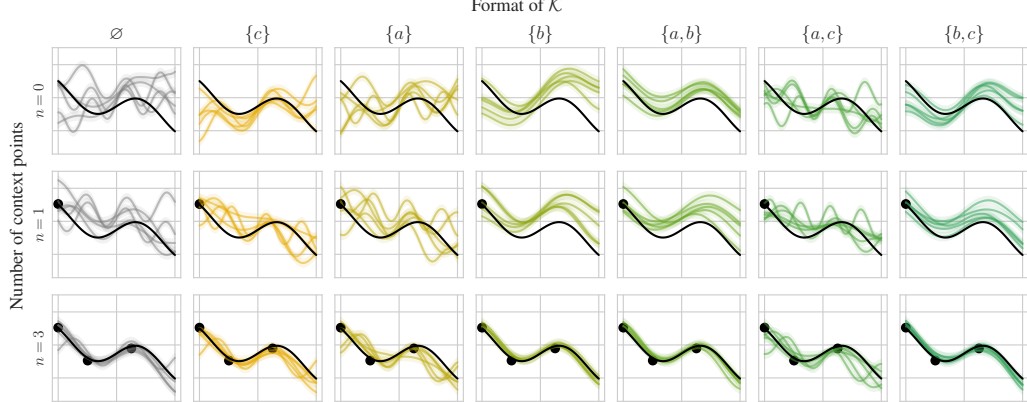

Figure A.1: *Sample predictions under varying formats of knowledge.* Knowledge about the value of the slope or frequency of oscillations provides global information about the overall shape of the function. Observing additional data points anchors the curves in the xy-coordinate system. Based on a qualitative investigation we conclude that the INP successfully learned how to integrate prior knowledge with observed data points.

### A.6.2 INFORMED WEATHER PREDICTIONS (SECTION 5.2.1)

We use the sub-hourly temperature dataset from the U.S. Climate Reference Network (USCRN)[2]. The data contains values of the air temperature measured at regular 5-minute intervals. For each task, the context and target datasets consist of measurements from one day. Training, validation, and testing collections of tasks are created by randomly selecting 507, 108, and 110 days, respectively, between the years 2021 and 2022 in Aleknagik, Alaska. For each task, the target dataset consists of all 288 measurements in the 24h range. Context observations are sampled by first uniformly sampling 10 data points and then selecting the chronologically first $n$ observations where $n \sim U[0, 10]$. We perform independent experiments with two formats of knowledge:

---

[2]https://www.ncei.noaa.gov/access/crn/qcdatasets.html

**A:** For each task, knowledge $\mathcal{K}$ is a vector encoding two values: the minimum temperature and the maximum temperature on the day. In this setup, the knowledge encoder, $h_{\theta_{e,K}}$, is a simple 2-layer MLP .

**B:** For each task, knowledge $\mathcal{K}$ is a synthetically generated "weather forecast" presented in a natural language format. For illustrative purposes, these weather descriptions were generated with GPT-4 OpenAI et al. (2023). In total, 726 descriptions, one per day were generated. The prompt used contains instructions to generate 2 sentences mimicking a weather forecast, based on 48 values sampled at 30-minute intervals from the ground truth temperature values. We use the following prompt:

```
System:  You are given a vector of values representing the temperature
for the next 24h at 30-minute intervals, starting at 12 am.  Your
task is to present the weather forecast according to these values.
Keep it to max 2 sentences.  Use descriptive words to refer to the
times of the day, e.g.  morning, afternoon, evening.

User: <<Temperature values>>
```

There are two reasons for using LLM-generated weather forecast instead of real ones: 1) availability of such data for research purposes not requiring web scraping; 2) the alignment of weather forecasts with the observed values available as our task-specific data. Real weather forecasts may contain a high degree of noise due to the predicted temperatures diverging from the recorded values. To learn a faithful mapping between the knowledge space and the function space that aligns with our human understanding, we require that information contained in the representations of knowledge is in agreement with the underlying data generating process. To ensure that the output of GPT-4 indeed aligns with the observed values, we have manually examined a random sample of the generated weather forecasts and found them to be of high quality.

In this setup, the knowledge encoder $h_{\theta_{e,K}}$ is implemented with a RoBERTa language model Liu et al. (2020) with all weights frozen except for the layer norm weights, which are tuned during the end-to-end training. The latent knowledge representation $k$ is obtained as a pooled sentence embedding. Here, we use the last hidden state of the CLS token.

For both setups A and B, the data encoder $h_{\theta_{e,D}}$ is implemented as a 3-layer MLP and the decoder $g$ as a 4-layer MLP. We used the MLP & FiLM aggregator $a$. We set the hidden dimension, $d = 128$. We use a learning rate of 1e-3 and set the batch size to 64. The knowledge representation is randomly masked at a rate 0.3 by setting $k = 0$. Vanilla NPs are known to underfit context observations and underestimate the variance, which became apparent with this more complex and noisy dataset. To mitigate this issue, in this experiment, we have employed multi-head cross-attention during the encoding of the data representation, $r$, as proposed by Kim et al. (2019). Precisely, $r = \sum_{i=1}^{n} \text{Att}_i h_{\theta_{e,D}}(x_i, y_i)$, where $\text{Att} = \text{MultiHead}(Q, K, V)$, with $Q$ being a matrix of target inputs, $K$ a matrix of context inputs and $V$ a matrix consisting of individual data representations $r_i = h_{\theta_{e,D}}(x_i, y_i)$. We use 4 attention heads.

See the main body of the paper for a discussion of the results. Figure A.2, shows sample tasks and their corresponding GPT-4 generated weather descriptions.

### A.6.3 FEW-SHOT AND ZERO-SHOT IMAGE CLASSIFICATION WITH CUB-200-2011 (SECTION 5.2.2)

We apply our model to zero and few-shot classification using the CUB-200-2011 dataset Wah et al. (2011). It contains 11,788 images of 200 subcategories belonging to birds. Following Akata et al. (2015), we use 100 bird categories for training, 50 for validation, and 50 for testing. We generate the labels for $N$-way classification tasks by choosing $N$ random classes at each training step and arbitrarily assigning the labels $0, \ldots, N-1$ to each. For each task, the number of shots $k$, i.e. the number of example images per class ranges uniformly between 0 and 10. The target set consists of 20 images per class. We perform independent experiments with three formats of knowledge:

**A:** Knowledge $\mathcal{K}$ represents attributes characteristic for a given class, e.g. wing span, feather color, shape of the beak. This is obtained by a class-wide average of the binary attribute vectors from

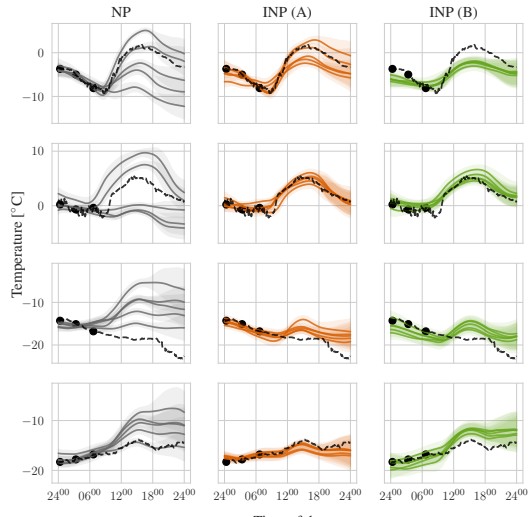

*The night will start off cold with temperatures falling to -8.9°C by late morning, and then gradually rise to a high of 1.6°C in the late afternoon. Temperatures will start to drop again in the evening, reaching -3.1°C by midnight.*

*The night will start off chilly with temperatures around 0.5°C, but it will drop to -1.7°C by early morning. The day will gradually warm up, reaching a high of 5.1°C in the afternoon before cooling off to 1.0°C by midnight.*

*The night will start off cold with temperatures falling to -16.8°C by dawn, and the day will continue to get colder, reaching a chilly -23.0°C by midnight. Afternoon temperatures will hover around -18.5°C, so bundle up if you're heading out.*

*The night will be bitterly cold with temperatures around -18 degrees, gradually increasing to -14 degrees by late afternoon. The temperature will slightly drop again to -15 degrees in the evening, warming up a bit to -14 degrees at midnight.*

Figure A.2: Sample predictions and the GPT-4 generated "weather forecasts" for setup B.

the original dataset associated with each image. Knowledge representations, $\mathcal{K}$, are constructed by stacking all $N$ class attribute vectors into a $N \times 312$ tensor. In this setup, the knowledge encoder, $h_{\theta_{e,K}}$, is a simple 2-layer MLP.

**B:** Knowledge $\mathcal{K}$ represents the average per class, natural language descriptions of the $N$ classes. These are obtained by averaging sentence embedding of individual image captions belonging to the given class. We use human-generated captions as collected in Reed et al. (2016) and encode them using CLIP embeddings Fu et al. (2022). Averaged per class text embeddings are then stacked to form a $N \times d^{model}$, where $d^{model} = 512$. In this setup, the knowledge encoder, $h_{\theta_{e,K}}$ is a 2-layer MLP.

**C:** We use GPT-4 to generate individual descriptions of each class based on the human-generated image captions. We present 5 randomly sampled image captions pertaining to one class and prompt GPT-4 to generate short descriptions of features characteristic of the given bird breed. To generate the class descriptions, we use the following prompt format:

```
System:   You are given 5 descriptions of a bird breed.  Based on this
information generate one comprehensive description of the bird
breed.  Keep it short and informative.
User: <<List of 5 randomly sampled image captions>>
```

In this setup, the knowledge encoder, $h_{\theta_{e,K}}$ is the CLIP text encoder. The embeddings of class descriptions are obtained as the average of all outputs from the last layer of CLIP. After stacking them together in a $N \times d^{model}$ tensor, they are passed through a linear projection layer.

For all setups, A, B, and C, the data encoder, $h_{\theta_{e,D}}$ is implemented with a frozen CLIP vision model, followed by a linear projection layer. Following the approach of Garnelo et al. (2018a), we only aggregate over inputs of the same class. The aggregated class-specific representations are then concatenated to form the final representation of size $N \times d$. We set $d = 512$. We use the sum & 2-layer MLP aggregation $a$. We modify the decoder to return the logits of the categorical distribution. For a $N$-way task with class labels $c_1, \ldots, c_N$, we define $p(y \mid x, z)$ as:

$$p_{\theta_d}(y = c_j \mid x, z) = \frac{\exp(-w_j^T x)}{\sum_{j'} \exp(-w_{j'}^T x)}, \quad [w_1, \ldots, w_N] = g_{\theta_d}(z), \; z \in \mathbb{R}^{N \times d},$$

where $x$ is a CLIP image embedding from the target set and $g_{\theta_d}$ is a 2-layer MLP. In our experiments, we use the Hugging Face implementation of the CLIP ViT-B/32 model

(https://huggingface.co/openai/clip-vit-base-patch32). We use a learning rate or 1e-4, batch size of 32 and knowledge is randomly masked at rate 0.5. For setups A and B, the INP model is trained end-to-end. For setups C, the weights of the INP model from the trained weights of the already trained, plain NP, and all model components, including the CLIP text encoder, are fine-tuned. As opposed to setup B, in setup C fine-tuning of the CLIP text encoder was necessary to ensure alignment between the class-wide descriptions and image representations. Empirically, the two-stage training resulted in improved convergence.

For the empirical results and short discussion, refer to the main body of the paper. In Table A.4 we present sample human-generated captions (used in setup B) and their corresponding GPT-generated class descriptions (used in setup C).

## A.7 UNCERTAINTY QUANTIFICATION WITH INPs

One particularly appealing property of Neural Process, which motivated their choice for the basis of our informed meta-learner, is the ability to sample from the posterior instead of returning a single point estimate. This allows us to measure the reduction in model uncertainty given prior expert knowledge and/or observed data. We are mostly interested in measuring the epistemic, rather than aleatoric uncertainty.

Aleatoric uncertainty refers to the notion of randomness seen as the variability in the outcomes which is due to inherently random, unpredictable effects. As opposed to this, epistemic uncertainty refers to uncertainty caused by the lack of knowledge about the true relationship between model inputs and outputs. By observing data, or by inserting prior knowledge into the model, the epistemic uncertainty is reduced.

A natural choice for measuring the epistemic uncertainty would be the (conditional) entropy. By comparing $\mathbb{H}[p(f)]$ with $\mathbb{H}[p(f \mid \mathcal{K})]$ or $\mathbb{H}[p(f \mid \mathcal{D}_C)]$ we can measure the impact of prior expert knowledge or observed data on the reduction in the epistemic uncertainty for a single learning task (for notational convenience, we omit the dependence of $p$ on $\theta$, making the expressions applicable both to the ground truth DGP and the approximate $p_\theta$). However, in INPs, we only have access to samples from the variational distribution and since the decoder is implemented as a neural network, evaluating the distribution over functions $f$ is not possible directly. Instead, we need to resolve to measure the uncertainty in the observation space. Thus, we are interested in computing

$$\mathbb{H}[p(y \mid x, I)], \quad I \in \{\mathcal{K}, \mathcal{D}_C, \mathcal{K} \cup \mathcal{D}_C, \varnothing\} \tag{17}$$

at a particular location $x \in \mathcal{X}$ in our input space, which can be then, for instance, averaged across uniformly distributed points in $\mathcal{X}$. The quantity in (17) is known as the predictive uncertainty. To approximate (17) for an input $x$, we rely on Monte-Carlo estimation by sampling $S$ functions based on our variational decoder.

$$\begin{aligned}
\mathbb{H}[p(y \mid x, I)] &:= -\int p(y \mid x, I) \log p(y \mid x, I) dy \\
&= -\int \left( \int p(y \mid x, f) p(f \mid I) df \right) \log \left( \int p(y \mid x, f) p(f \mid I) df \right) dy \\
&\approx -\int \left( \frac{1}{S} \sum_{s=1}^{S} p(y \mid x, f^{(s)}) \right) \log \left( \frac{1}{S} \sum_{s=1}^{S} p(y \mid x, f^{(s)}) \right) dy
\end{aligned} \tag{18}$$

For each sample $f^{(s)}$, $p(y \mid x, f^{(s)})$ has a closed-form expression–in the case of regression it is modelled with a normal distribution. Thus, (18) can be computed by numerically approximating the integral in the last line. Note that, since predictive uncertainty is measured in the observation space, it also encompasses the uncertainty associated with the observational noise. Depeweg et al. (2018) suggest that (17) can be decomposed as:

$$\mathbb{H}[p(y \mid x, I)] = \underbrace{\mathbb{I}(y, f \mid x, I)}_{\text{epistemic}} + \underbrace{\mathbb{E}_{f \sim p(f \mid I)}[\mathbb{H}[p(y \mid x, f)]]}_{\text{aleatoric}} \tag{19}$$

The second part, $\mathbb{E}_{f \sim p(f \mid I)}[\mathbb{H}[p(y \mid x, f)]]$, is the average entropy when the predictive function is known, thus can be interpreted as the aleatoric uncertainty. If we model $p(y \mid x, f)$ with a

Table A.4: Example images, image captions and GPT-generated class descriptions.

| Sample Images | Sample image captions | GPT-generated class description |
|---|---|---|
| 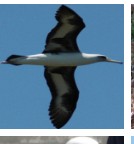 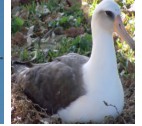 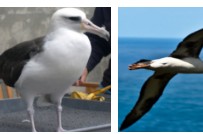 | 1. *A large bird with a white belly, black and white wings with a long beak.*
2. *This bird is white and grey in colour with a curved beak, and black eye rings.*
3. *A large bird with a white belly and face, black back and wings, and peach bill.*
4. *Bird has gray body feathers, white breast feather, and long beak*
5. *A medium sized bird with black wings, and a bill that curves downwards* | *This bird breed is a medium to large size, characterised by its grey body feathers, contrasting white belly and face, black back and wings, distinctive black eye rings, and a long, downward-curving peach bill.* |
| 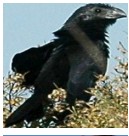 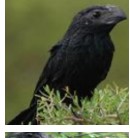 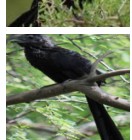 | 1. *This big bird has a sharp beak and has black covering its body.*
2. *An all black bird with a distinct thick, rounded bill.*
3. *This entirely black bird has long and wide rectrices relative to the size of its body.*
4. *A black bird with a long tail and large beak.*
5. *This black bird has sparse plumage and a thick brown beak.* | *This bird breed is large and entirely black with sparse plumage, characterised by its thick brown beak, long tail, and wide rectrices relative to its body size.* |
| 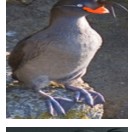 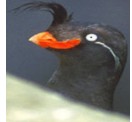 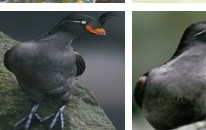 | 1. *This goofy looking bird sports webbed feet and a bright orange bill, with piercing white eyes and a dull coat of gray.*
2. *A black bird with a small, orange beak and a inverted feather curl at the base of the beak.*
3. *A black body, white eye with stripe next to it, and an orange bill are on this bird.*
4. *This black bird has a orange bill with hair coming out of it, small pupils, and a white line across its face.*
5. *This bird has wings that are black and has an orange bill* | *This bird breed is characterised by its black body, webbed feet, a bright orange bill with an inverted feather curl at the base, piercing white eyes with a distinctive stripe, and a dull grey coat.* |
| 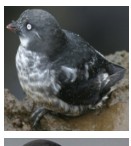 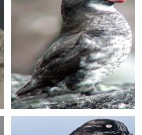 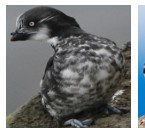 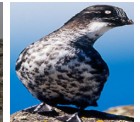 | 1. *This is a black bird with a white spotted belly and a white eye.*
2. *This bird is black with white and has a very short beak.*
3. *This bird has wings that are black and white and has a small bill*
4. *This small bird is white with black spots, a white neck, and black around its eyes.*
5. *This is a short stocky bird with webbed feet, it is mostly white with black wings and black speckles throughout.* | *This bird breed features a black body with a white and black spotted underbelly, a white and grey speckled chest, a black crown, bright white eyes with very small pupils, and a short, pointed, black and orange bill.* |

normal distribution, $\mathbb{H}[p(y \mid x, f)]$ has a closed-form expression, $\frac{1}{2}\log(\sigma(x)2\pi e)$, where $\sigma^2(x)$ is the variance at location $x$.

The first part, $\mathbb{I}(y, f \mid x, I)$, representing the information gain can be interpreted as the epistemic uncertainty of interest. This quantity can be computed as the difference of $\mathbb{H}[p(y \mid x, I)]$ and $\mathbb{E}_{f \sim p(f|I)}[\mathbb{H}[p(y \mid x, f)]]$, where both quantities are easy to estimate, as discussed above.

## A.8  ADDITIONAL EXPERIMENTS

### A.8.1  MODEL PERFORMANCE VS. NUMBER OF TRAINING TASKS

**Setup:** We follow the same setup as in the illustrative experiment from section 5.1.1. We create multiple training collection of tasks with a varying number of total training tasks, $N^{train} \in \{25, 50, 75, 100, 200, 1000\}$, with the upper limit being the number of tasks used in the original experiment. For each training collection of tasks, we train independent INP and NP models. The INP models receive information about two, one or none of the parameters $a$, $b$ or $c$ via knowledge representations, $\mathcal{K}$. All models are validated and tested on the same collection of validation / testing tasks.

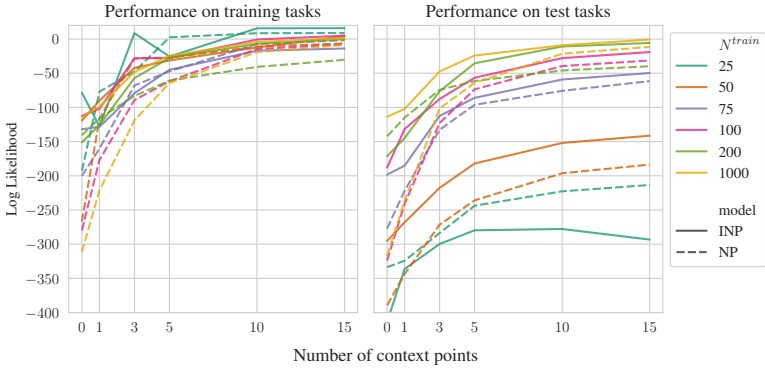

Figure A.3: Log likelihood of target data vs. number of context data points (higher is better). Comparison across varying number of all training tasks, $N^{train}$. Left - model performance on training tasks, Right - model performance on test tasks.

Figure A.3 shows the performance of all models on training (left) and testing (right) tasks. We observe that: 1) Both for the NP and INP models, as the number of training tasks decreases the performance gap between training and testing tasks increases. We note that this performance gap is already at a (subjectively) reasonable level with only as few as 75 training tasks. 2) For all INP models trained with $N^{train} \geq 50$ tasks we the additional knowledge presented for each task improves the performance over the plain, uninformed NP. When the number of training tasks is too small, here $N^{train} = 25$, we observe a "knowledge overfitting" effect. With insufficient number of training tasks the INP is unable to appropriately capture the relationship between knowledge and empirical data, and thus fails to generalise to new, previously unseen tasks and their corresponding, also previously unseen, knowledge representations.

**Take-away:** We tested the robustness of the INP model to the reduction in the number of training tasks. We showed that in the experimental setup of section 5.1.1, adding external knowledge continues to deliver noticeable performance gains over the uninformed NP when dropping from 1000 to as few as 50 training tasks. We also noted that with too few training tasks, the INP may fail to generalise. To prevent this effect from occurring in real-world deployment, we advise testing the model on held-out validation tasks and comparing its performance against an uninformed baseline, monitoring the knowledge overfitting effect. We note that in real-world applications, more training tasks can be obtained by (semi-)synthetic data generation, augmenting the size of the meta-training set to improve model generalisation.

A.8.2  MODEL PERFORMANCE AND KNOWLEDGE COMPLEXITY

**Setup.** To assess the impact of knowledge complexity on the efficacy of learning the relationship between knowledge and the model hypothesis space, we again follow the same setup as in section 5.1.1. We create multiple training collection of tasks with a varying number of total training tasks, $N^{train} \in \{25, 50, 75, 100, 200, 1000\}$. All models are validated and tested on the same collection of validation / testing tasks. For each setting of $N^{train}$ we train an uninformed NP and 3 independent INP models with different knowledge representations used during training:

- $\mathcal{M}_{abc}$ is a model where for each task its corresponding knowledge encodes, at random, one of the three parameters, $a$, $b$, or $c$;
- $\mathcal{M}_{ab}$ is a model where for each task its corresponding knowledge encodes, at random, one of the two parameters: $a$ or $b$ (the value of $c$ is never revealed);
- $\mathcal{M}_{b}$ is a model where for each task its corresponding knowledge encodes the value of $a$ (the values of parameters $a$ and $c$ are never revealed).

Knowledge representations are constructed by one-hot encoding the type of the revealed parameter with its value appended at the end. We note that for the INP models $\mathcal{M}_{b}$, $\mathcal{M}_{ab}$, $\mathcal{M}_{abc}$, the complexity of knowledge representations gradually increases; the knowledge space is 1, 2 and 3 dimensional, respectively. We hypothesise that as the complexity of the knowledge space grows, more training tasks are needed to effectively learn the mapping from knowledge representations to prior distributions over functions. Given the same number of training tasks, the INP model $\mathcal{M}_{ab}$ needs to learn how to disentangle the information about the function's oscillations (parameter $b$) from the information about the function's slope (parameter $a$). Model $\mathcal{M}_{abc}$ additionally needs to discover the meaning of knowledge about the intercept (parameter $c$). Therefore, we expect that, given *the same number of context points* and *the same information contained in $\mathcal{K}$*, the relative performance gains of the INP models $\mathcal{M}_{b}$, $\mathcal{M}_{ab}$, $\mathcal{M}_{abc}$ over the uniformed NP model should decrease as the complexity of knowledge space increases.

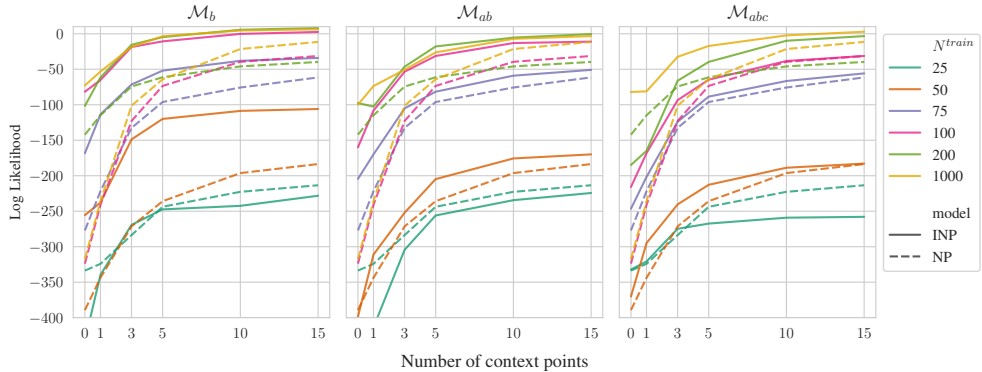

Figure A.4: Log-Likelihood of target data vs. number of context data points (higher is better). Comparison across a varying number of tasks used for training $N^{train}$. Complexity of knowledge space grows from left to right. All INP models are presented with the same knowledge about each task–the value of the parameter $b$.

Figure A.4 shows the log-likelihood of the target data evaluated on 500 testing tasks. For every INP model at test time we reveal the same information via knowledge representations—the value of the parameter $b$. Firstly, we observe the same two effects as in experiment A.8.1. With more training tasks, model performance improves. 2) An insufficient number of training tasks may lead to the "knowledge overfitting" effect; here at $N^{train} = 25$ the INP performs worse than the NP. Secondly, we look at the performance gap between the INP and the NP (the gap between solid and dashed lines). We observe that as the complexity of the knowledge space grows (left to right) the performance gap between the INP and the NP decreases. This is summarised through the $\Delta$AUC metric, presented in the Table A.5. From Figure A.4 we can also conclude that the more complex the the knowledge space is the more training tasks are needed to effectively train an INP model. For instance, performance of the INP model $\mathcal{M}_{b}$ trained with $N^{train} = 75$ tasks is comparable to the performance of the INP $\mathcal{M}_{abc}$ trained with $N^{train} = 100$ tasks.

Table A.5: Average relative performance improvement (%) between informed and uninformed predictions. The performance gains become smaller as the complexity of the knowledge space grows (top to bottom). $N^{train}$ is the number of training tasks used. INP performs better than the NP for all settings of $N^{train} \geq 50$ indicating effective transfer between knowledge representations and functional priors. INP overfits with not enough training tasks, here at $N^{train} = 25$.

| $N^{train}$ model | 1000 | 500 | 200 | 100 | 75 | 50 | 25 |
|---|---|---|---|---|---|---|---|
| $\mathcal{M}_b$ | 81.74 | 83.51 | 64.52 | 79.1 | 44.78 | 39.71 | -8.52 |
| $\mathcal{M}_{ab}$ | 65.3 | 67.45 | 43.46 | 54.66 | 22.41 | 6.67 | -18.67 |
| $\mathcal{M}_{abc}$ | 71.76 | 57.51 | 2.02 | 26.46 | 9.48 | 8.05 | -5.6 |

**Take-away:** The above experiment confirms our hypothesis about the complexity of the information conveyed in knowledge representations and the hardness of learning the mapping between knowledge representations and the model hypothesis space. As the complexity of knowledge increases, more training tasks are needed to effectively learn the relationship between knowledge representations and the functional priors.

### A.8.3 Correlation in training data and knowledge disentanglement

**Setup:** For each task, context and target data points are sampled according to a similar process as in the experiments from section 5.1.1. A function $f$ is sampled from the family of sinusoidal functions with a linear trend, $f(x) = ax + \sin(bx)$. As previously, we also introduce a Gaussian observational noise, s.t. $y_i = f(x_i) + \epsilon_i$, $\epsilon_i \sim \mathcal{N}(0, 0.2)$. In this experiment. we simulate a scenario in which the training data exhibits a potentially spurious correlation. We sample the parameters $a$ and $b$ from a multivariate Gaussian,

$$\begin{bmatrix} a \\ b \end{bmatrix} \sim \mathcal{N} \left( \begin{bmatrix} 0 \\ 3 \end{bmatrix}, \begin{bmatrix} 1 & \sigma \\ \sigma & 2 \end{bmatrix} \right)$$

We create 6 training and validation collection of tasks, one for each value of the covariance between $a$ and $b$, $\sigma \in \{0.0, 0.3, 0.6, 0.9, 1.2, 1.4\}$. We then train 6 independent INP and NP models. For the INP models we let $\mathcal{K}$ encode the value of one of the two parameters $a$ or $b$. The number of context points $n$ ranges uniformly between 0 and 10; the number of targets is set to $m = 100$. The testing collection of tasks is created by sampling functions where $a$ and $b$ are *independent* (i.e. $\sigma = 0.0$). This setup aims to test the robustness of the INP model to spurious correlations in the training data. We want to investigate whether the INP model is able disentangle the meanings of parameters $a$ and $b$. In this setup, $p(\mathcal{K}|f)$ remains the same at training at test time–it always provides partial information about the value of one of the parameters. It is the underlying process $p(f)$ that changes between training and test time.

Results presented in table A.6 show that when the correlation between the parameter $a$ and $b$ increases, the test-time performance of both the INP and NP models downgrades. This is due to the train-test distribution shift. Moreover, when the correlation is moderate ($\rho \leq 0.64$), the INP model outperforms or mathces the performance of the NP. We note, however, that for $\rho >= 0.42$, the zero-shot predictions ($n = 0$) are better for the uninformed model than the INP. This is also true for all values of $n$ at higher correlation levels ($\rho \geq 0.85$). We hypothesize that this is because the INP has overfitted to the correlation between the parameters $a$ and $b$. In the training dataset, revealing the information about the value of one parameter gives information about the value of the other, unrevealed parameter. INP exploits this dependency.

**Take-away:** INPs learn the meaning of knowledge based on its relationship with the empirical data. If this relationship changes at test time, good performance of the INP can no longer be guaranteed. This characteristic may be especially dangerous when there are spurious correlations in the dataset. The INP is prone to overfitting to these correlations, "misunderstanding" the true meaning of knowledge, and thus failing to generalise to new knowledge representations and their corresponding tasks, where the spurious correlations are no longer present.

### A.8.4 Meta-learned vs. exact knowledge integration

Table A.6: Average log-likelihood on test tasks vs. correlation in training data (higher is better). $\rho$ - the correlation coefficient between random parameters $a$ and $b$. $n$ - number of context data points per task. Model results for which the log likelihood is higher by a statistically significant margin highlighted in bold. Values in brackets stand standard errors estimated with bootstrap.

| $\rho$ | 0.00 | | 0.21 | | 0.42 | |
|---|---|---|---|---|---|---|
| model | INP | NP | INP | NP | INP | NP |
| $n = 0$ | **-139.1 (10.2)** | -209.2 (8.3) | **-174.4 (13.7)** | -196.5 (7.7) | -266.1 (19.6) | **-221.7 (9.2)** |
| $n = 1$ | **-99.0 (10.6)** | -102.1 (6.5) | **-73.1 (6.2)** | -120.4 (5.4) | **-95.0 (9.4)** | -108.6 (4.4) |
| $n = 4$ | **-16.9 (1.7)** | -30.9 (2.6) | **-34.8 (3.9)** | -41.0 (3.4) | **-29.2 (3.8)** | -38.2 (3.4) |
| $n = 5$ | -12.1 (2.0) | -11.9 (2.0) | -15.7 (1.8) | -17.8 (2.6) | **-12.4 (2.5)** | -21.7 (2.9) |
| $n = 10$ | 1.3 (1.1) | 1.8 (1.4) | -0.5 (1.2) | 2.1 (0.9) | **0.2 (0.9)** | -4.5 (2.0) |
| $n = 15$ | 3.5 (0.7) | 5.4 (1.4) | 0.8 (0.8) | 2.6 (1.6) | **3.1 (0.6)** | -2.2 (2.1) |

| $\rho$ | 0.64 | | 0.85 | | 0.99 | |
|---|---|---|---|---|---|---|
| model | INP | NP | INP | NP | INP | NP |
| $n = 0$ | -356.4 (22.2) | **-214.0 (9.2)** | -795.6 (38.9) | **-321.8 (14.8)** | -1367.3 (63.6) | **-410.7 (15.2)** |
| $n = 1$ | **-108.1 (7.1)** | -160.3 (7.4) | -234.4 (9.3) | **-200.0 (9.9)** | -830.9 (40.9) | **-527.3 (19.2)** |
| $n = 3$ | **-26.2 (2.2)** | -64.6 (4.4) | -149.0 (6.6) | **-118.6 (6.3)** | -551.5 (32.5) | **-360.1 (10.6)** |
| $n = 5$ | **-18.3 (2.0)** | -30.3 (3.0) | -101.4 (4.9) | **-94.0 (5.1)** | -404.2 (13.2) | **-319.9 (9.0)** |
| $n = 10$ | **-6.2 (1.2)** | -18.2 (2.6) | -81.2 (4.4) | **-70.6 (4.5)** | -342.7 (11.1) | **-324.9 (9.3)** |
| $n = 15$ | **-4.3 (1.2)** | -11.6 (2.3) | -74.1 (4.3) | **-66.2 (4.3)** | -332.2 (11.1) | **-313.0 (8.9)** |

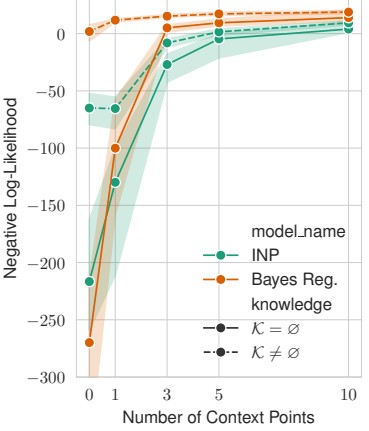

Figure A.5: *Knowledge integration via the INP vs. exact inference with Bayesian regression.*

The primary advantage of informed meta-learning lies in its universality and applicability to scenarios where model-based approaches to knowledge integration are infeasible or require significant human efforts. Unlike informed meta-learning, model-based approaches require knowledge representations to be easily translatable into explicit inductive biases, such as functional representations, parameter priors, or regularisers for the loss function. On the other hand, a model-based approach may guarantee the correctness of knowledge integration by design instead of relying on a neural approximation learned based on a finite number of meta-training tasks. However, the universality of informed meta-learning necessarily comes with a trade-off in performance. In this section we investigate the efficacy of the data-driven approach of informed meta-learning to knowledge integration in comparison to a model-based approach–fitting an explicit parametric model encoding the available task-specific knowledge exactly.

**Setup.** We again follow the setup of section 5.1 with sinusoidal functions. We compare the INP and NP models, against an exact Bayesian regression model with observations modelled according to: $p(y|x) = \mathcal{N}(y; \mu(x), \sigma^2)$, where $\mu(x) = ax + \sin(bx) + c$ and we put priors on the parameters $a$, $b$, and $c$ that correspond precisely to the ground-truth process $p(f)$ used in your setup, i.e. $a \sim U[-1, 1]$, $b \sim U[0, 6]$, $c \sim U[-1, 1]$. We fix $\sigma$ to its ground-truth value of 0.2. For a given task with knowledge $\mathcal{K}$ about the value of one or two of the parameters $a$, $b$, $c$, we fix the value of the respective parameters and only estimate the posteriors $p(\cdot|\mathcal{D}_C)$ for the remaining, unknown quantities. Given the non-linear transformation of the parameter $b$ through the sine function, we find the posterior of the parameters $a$, $b$, $c$ with MCMC estimation with the NUTS algorithm. For each task, we run 4 chains with 1000 burn-in samples and 2000 samples used for estimation.

**Results.** Figure A.5 compares the performance of predictions made based according to an exact regression model and the meta-learned predictive posterior with INPs. We first note that already for the the uninformed case, $\mathcal{K} = \varnothing$, the INP performs worse than the Bayesian regression model, specifically when more data becomes available. This is expected, as the regression model is ex-

plicitly based on the ground-truth data generating process Meanwhile, the Neural Process only approximates this distribution over sample functions by learning on a finite collection of meta-training tasks and is also known to underfit contextual observations. In terms of knowledge integration, we observe that even thought the informed predictions of the INP significantly improve upon the non-informed INP (which is effectively the plain NP), there is still a noticeable gap between the INP and the theoretically achievable upper bound dictated by the informed Bayes regression. It is also worth comparing the inference times of Bayes regression with MCMC estimation vs. that of the (I)NPs. After training, predictions with (I)NPs simplify to a forward pass through the trained neural network, which for a single task in this instance takes less than 0.004 seconds on a machine with the AMD Epyc Milan 7713 CPU, 120GB RAM and the NVIDIA 48GB A6000 GPU. In comparison, estimating the parameters of the regression model with MCMC took us on average 8.6s per task. For reference, training an INP model in this instance takes around 20min.

**Take-away.** The major advantage of informed meta-learning is its applicability across diverse knowledge and data formats, as illustrated with the real-world and natural language data experiments of section 5.2. However, as demonstrated with the above experiment, this universality comes with trade-offs in performance. When a model-based approach is available, it should be preferred, as it may guarantee the correctness and exactness of knowledge integration by design instead of relying on a neural approximation learned based on a finite number of meta-training tasks. The appeal of informed meta-learning becomes evident when the underlying functional form of the data generating processes is not known explicitly and the task-specific knowledge is difficult to represent in a form of an exact mathematical equation or a constraint. In such cases, model-based approaches to knowledge integration become challenging if not infeasible.

