# OpenReview forum: "Towards Automated Knowledge Integration From Human-Interpretable Representations"
_ICLR.cc/2025/Conference — ICLR 2025 Spotlight_

### Official Review · Reviewer_gpLz · 2024-11-03

**Soundness:** 3
**Presentation:** 3
**Contribution:** 3
**Rating:** 8
**Confidence:** 3

**Summary:**

This paper studies a meta-learning setup in which a learner tries to adapt to a new environment given only a few samples.

The paper's main idea is to endow the learner with additional domain knowledge about the specific task (\mathcal{K} in the paper).

The key research question is whether, in the era of LLMs, we can provide the learner with human-interpretable domain knowledge in the form of text, e.g. the human provides the learner with the information that the function "is monotonically decreasing".

The paper first develops a formal framework for meta-learning with additional information (Section 3).

The paper then provides different empirical applications of the new paradigm using Neural Processes (Section 5).

The paper states that *"our work primarily focuses on putting forward the idea of informed meta-learning with the introduced
class of INPs serving mainly as an illustration."*

Overall, this is an interesting and well-written paper with limited empirical evaluation (which makes the currently submitted paper a marginal paper in my view).

I'm not very familiar with the meta-learning literature, and I decided to give this paper the benefit of the doubt with a score of 6.

**Strengths:**

I find the research question quite interesting. It connects a classic topic in machine learning research—how to integrate human domain knowledge into the algorithm's inductive bias—with recent developments (LLMs).

The paper is well-written, and the appendix documents the different experiments in detail. The paper illustrates key ideas and results with well-designed figures.

The paper's theory seems sound (though I did not check the details), and the experiments are well-designed.

Overall, this paper makes some interesting suggestions for how novel developments in LLMs can inform existing work in knowledge integration / meta-learning.

**Weaknesses:**

The paper's main weakness is the empirical applications. Specifically, it does not provide a real-world application in which providing additional information in natural language improves upon other approaches (running somewhat contrary to the motivation as outlined, for example, in Figure 3).

The paper is aware of this and states that *"The aim of our work is not to present a new method that surpasses existing baselines on a benchmark dataset; rather, we propose a new viewpoint on meta-learning as a means of establishing an interface
between human domain knowledge and the hypothesis space of ML models."*

However, I'm uncertain how interested the ICLR audience will be in the methodological approach alone.

**Section 5.1:** The experiments in section 5.1 are all for the synthetic function y_i = ax_i + sin(b x_i) + c + epsilon_i. For this reason, the results in this section (while convincing) read like a case study, not a systematic assessment of the pro's and con's of the proposed method.

**Missing baselines:** I find the usefulness of the proposed method hard to gauge because the paper does not compare it with other existing methods. For example, the authors mention that Bayesian linear regression would be a natural method to use in the setup of Section 5.1. Then why does the paper not compare performance with Bayesian linear regression?

**Questions:**

**Question 1:** Figure 3 contains nice examples, e.g. K = "This function is sharply decreasing." However, there seems to be no exmpirical application with K = natural language for a regression problem in the paper. Why not? Did the authors not try this, or did they try it, and it did not work? I would be quite interested to see if this can work well in practice.

**Question 2:** Figure 4(b): I do not understand how the plot value for {a,b} can be in between {a} and {b}? should {a,b} not be strictly better than {a} and {b}?

**Question 3:** What exactly are the research contributions in Section 3? To me, this Section mostly reads like a summarization of different possible approaches, laying out the strategy the authors want to employ in the paper.

**Question 4:** Concerning the general motivation, it is a bit strange to read some of this in 2024. For example, from the introduction: *"with the knowledge integration step often forming the core contribution of many ML papers"*. Do the authors actually believe that "the knowledge integration step" will be a core part of many ICLR'25 papers?

**References:** Some of the references need updating. e.g.

"Chelsea Finn, Kelvin Xu, and Sergey Levine. Probabilistic Model-Agnostic Meta-Learning, 2019.
eprint: 1806.02817."

was published at neurips 2018.

---

> ### Author Response · Authors · 2024-11-23
> **Response to your review -- part 1**
>
> We thank the reviewer for their insightful comments and constructive feedback. Below, we address the questions and concerns raised by the reviewer.
>
> ---
>
> ### Weaknesses
>
> **W1) Practical applications.**
>
> We would like to highlight to the reviewer that our empirical evaluation does, in fact, contain two experiments with real-world data and additional information in natural language.
>
> Section 5.2.1 (setup B) presents an experiment on regression tasks with knowledge representations in a natural language format describing the evolution of the temperature throughout a given day. The natural language descriptions in this experiment resemble those presented in Figure 3, but at the same time are longer, more complex, and the language is specific to the domain of weather forecasting. They include statements like : “temperatures will drop to -5 degrees by early morning”, “ the temperature will gradually increase from 2 to 18”, etc. (see Figure. A.2. for more).
>
> Section 5.2.2 (setup C) presents an experiment on image classification tasks with knowledge represented as textual descriptions of the characteristic features for each bird species that we aim to classify.
>
> Both experiments highlight that providing the additional information significantly improves upon the non-informed predictions. The aim of these experiments is not only to validate the empirical effectiveness of our approach but also to illustrate its potential in practical, knowledge-rich domains where natural language insights may improve low-data learning.
>
>  We can draw parallels between these experiments and other domains. To inspire future applied research we have provided additional example applications of informed meta-learning in the below table:
>
> | Domain | Task type | Data | Knowledge |
> | --- | --- | --- | --- |
> | Finance | Economic or stock market forecasting | GDP over time, or daily stock prices | Descriptions of trends e.g. *”The stock prices for APPL are expected to continue the upward trend”*. |
> | Healthcare | Disease classification | Patient health measurements | Expected relationships between symptoms and diseases, e.g., *“High blood pressure is often linked to cardiovascular issues.”* |
> | Medical Imaging | Disease detection in X-rays |  Images of lungs  | Relationships between abnormalities and diseases, e.g., *"Presence of shadows in upper lung areas may suggest tuberculosis."* |
> | Energy | Demand forecasting | Hourly electricity consumption data | Expectations based on events or trends, e.g., *“Energy demand is expected to spike in winter months due to heating needs.”* |
> | Logistics | Delivery time estimation | Delivery routes, traffic patterns | Expected effects of traffic and weather, e.g., *“Deliveries are typically delayed during peak hours in urban areas.”* |
>  | Retail | Demand prediction | Sales data, time series | Expected shopping behaviours, e.g., *“Sales tend to increase during holiday seasons, especially for electronic goods.”* |
>
>  **UPDATE:** We propose to include the above table in the Appendix of our paper to inspire future applications of informed meta-learning in the real-world.
>
> **W2) The purpose of Section 5.1**
>
> The purpose of this section is to demonstrate empirically the performance of the informed meta-learning framework as instantiated through the method of INPs in a controlled and fully tractable environment, with no influence of confounding factors that may be present in real-world data. This section also introduces the key metrics used in the subsequent real-world data experiments of section 5.2. While we acknowledge the toy nature of this section, we nevertheless find it useful for building the reader’s intuition, understanding of the method and the training / evaluation pipelines.  In this treatment, we follow the convention of the original works on Neural Processes (Garnelo et al., 2018), in which the first introductory experiment is fitted to data sampled from a well-defined stochastic process with subsequent experiments conducted on real-world data generated according the an unknown DGP.

---

> ### Author Response · Authors · 2024-11-23
> **Response to your review -- part 2**
>
> **W3) Lack of comparison to model-based approaches**
>
> Thank you for your comment. We would like to highlight that the primary advantage of informed meta-learning lies in its universality and applicability to scenarios where model-based approaches to knowledge integration are infeasible or require significant human effort. Our experiments are designed to demonstrate the feasibility of knowledge integration via informed meta-learning, and we compare INPs against the non-informed baseline of NPs to validate the effectiveness of meta-learned, knowledge-dependent priors.
>
> Unlike informed meta-learning, model-based approaches require knowledge representations to be easily translatable into explicit inductive biases, such as functional representations, parameter priors, or regularisers for the loss function. In the specific case of Section 5.1, such a model-based approach is indeed feasible: it is easier to fit an explicit regression model with fixed parameter values derived from $\mathcal{K}$’s  than to learn the relationship between $\mathcal{K}$’s and the neural network’s parameters from data. Consequently, Bayesian linear regression would naturally outperform INPs in this well-defined environment of tasks. This scenario, however, is not representative of the expected use case of informed meta-learning, in which the underlying data-generating processes are unknown and translation of knowledge into explicit inductive biases is challenging.
>
> In Section 5.2, we demonstrate the usage of informed meta-learning with real-world time series and image data and where knowledge is represented in natural language. In these cases, model-based approaches to knowledge integration are inapplicable. To the best of our knowledge, informed meta-learning is the only feasible method for integrating knowledge irrespective of its format or its relationship to the ground truth DGP.
>
> In summary, the major advantage of informed meta-learning it its versatility and universality across diverse scenarios, knowledge, and data formats. That said, we acknowledge that this universality comes with trade-offs in performance: when a model-based approach is available, it should be preferred, as it guarantees the correctness of knowledge integration by design instead of relying on a neural approximation learned based on a finite number of meta-training tasks. We highlight this trade off in the final section of the paper (lines 510-517).
>
> ### Questions
>
> **Q1) *There seems to be no empirical application with K = natural language for a regression problem in the paper.***
>
> We would like to note that the temperature prediction experiment presented in section 5.2.1., setup B, is precisely a regression problem where knowledge is presented in natural language. The presented sentence describe the evolution of the temperature curve throughout the day, including expressions like *“temperatures will drop to -5 degrees by early morning”*,  *"the temperature will gradually increase from 2 to 18”*, etc. (see Figure. A.2. for sample knowledge representations).  These kinds of descriptions are in a close resemblance to the ones presented in Figure 3. In fact, they are even more complex and detailed.

---

> ### Author Response · Authors · 2024-11-23
> **Response to your review -- part 3**
>
> **Q2) Figure 4(b): I do not understand how the plot value for {a,b} can be in between {a} and {b}? should {a,b} not be strictly better than {a} and {b}?**
>
> Thank you for pointing this out. Indeed, it is expected that upon successful knowledge integration, providing more knowledge (here knowledge about parameters a an b as opposed to just a or just b), should result in better predictions. In Figure 4b), the improvement for $\mathcal{K} \sim \\{b\\}$ and $\mathcal{K} \sim \\{a\\}$ is within the standard error of the improvement when $\mathcal{K} \sim \\{a, b\\}$, meaning the difference is not statistically significant. The reported mean improvement is sensitive to outliers as the variance of relative values is generally higher than that of absolute values. In addition to providing average relative $\Delta \text{AUC}$ across testing tasks, we include the average values of the negative log-likelihood across diferent knowledge types in Table A.2 in the Appendix. Comparison of the average loss across tasks agrees with our expectations: providing knowledge about 2 parameters results in better predictions than just a single parameter. Although for the case of $\{b\}$ and $\{a, b\}$ these differences are not significant—this is may be caused by the fact that the value of  $b$ combined with a 1 or 2 datapoints almost fully determines the shape of the underlying curve, making the learning of the relationships between knowledge of type $\\{a, b\\}$  challenging, as for tasks with $\mathcal{D}_C \neq \varnothing$, the value of $a$ may appear redundant for the model.
>
> We also note that, in practice, the learned prior $p_\theta(f \vert \mathcal{K})$, and the posterior predictive $p_\theta(y \vert x, \mathcal{D}_C, \mathcal{K})$ , are only a neural approximation of the true, underlying distribution, as learned based on the finite meta-training set ot tasks. The quality of this approximation will depend on the the number of meta-training tasks, the train-test distribution shift, and the chosen model hyperparameters (see the remarks section of the paper). Thus, in practice, we may observe empirical results deviating from the theoretically optimal behaviours. In this instance, while providing knowledge about both parameters $a$ and $b$ should be better on average across all tasks, we may find individual tasks in which improvement will is not strictly better than just providing the value of a single parameter. This is one of the disadvantages of using meta-learned priors instead of conventional model-based approaches, in which accurate knowledge integration can be guaranteed by design. We make this point in the Limitations section of our paper and in the analysis of the results from section 5.2.1 (lines 449-454).
>
> **Q3) Research contributions of Section 3.**
>
> The research contributions of this section are as follows:
>
> - Section 3.2: Formally defining and casting knowledge-conditioned meta learning as a way of performing automated knowledge integration. This view point on meta-learning is novel.
> - Examples 1 and 2 present possible ideas for concrete implementation of informed meta-learners, laying our foundations for future work and the later introduced method of INPs.
> - Section 3.2.1. motivates theoretically the proposed approach.
> - Section 3.2.1. highlights the challenges associated with practical implementation of informed meta-learners pointing at aspects worth paying attention to during empirical evaluation of informed meta-learners (this section is tightly linked with the questions investigated in the empirical study of section 5 and the additional experiments presented in section 5). It also points at the directions for future work.

---

> ### Author Response · Authors · 2024-11-23
> **Response to. your review -- part 4**
>
> **Q4) The paper's motivation and relevance to ICLR.**
>
> Thank you for this comment. Knowledge integration indeed remains an important area of research; its success is critical for applications in low-sample and noisy data environments, as noted in our introduction. However, we do not expect many ICLR 2025 paper to focus on this topic, given that conventional knowledge integration methods are often developed in a highly application-specific manner. Such methods are tightly bound to particular knowledge formats and data domains, which limits their relevance for broader ML research audiences.
>
> This is, however, in a sharp contrast to our presented approach of informed meta-learning, which addresses these limitations by offering a flexible approach to knowledge integration.  Unlike conventional methods, the informed meta-learning framework is agnostic to the format of knowledge representations and their relationship with the observable data. This makes it applicable across a wide range of domains, where expert knowledge may be present, such as healthcare, finance, or environmental science. We therefore believe that our paper is relevant and of interest to the broader ICLR audience.
>
>
> **Q5) Updating the references**
>
> Many thanks for pointing this out. **UPDATE:** Where applicable, we have now updated all references from preprints to their corresponding conference / journal publications.
>
> ---
>
> We greatly appreciate the time and effort you’ve dedicated to reviewing our paper. We hope that the resulting additional experiments will strengthen the contributions of this paper and that our responses address your concerns satisfactorily.
>
> Thank you again for your valuable insights and for helping us improve the quality of this work.
>
> The authors of submission #11605

---

> > ### Comment · Reviewer_gpLz · 2024-11-24
> > **Thanks for the author response**
> >
> > Thanks for the detailed author response. Based on reading the other reviews and the rebuttal, my current impression is that this is a good paper. I will discuss it further with the other reviewers before I decide on my final score.
> >
> > Two more comments:
> >
> >
> > **" Consequently, Bayesian linear regression would naturally outperform INPs in this well-defined environment of tasks. "**
> >
> > I agree with the author's response, but I believe the reader would still be interested to see the performance of INPs vs. Bayesian linear regression in this setup.
> >
> >
> > **Why not use real weather forecasts? (Reviewer EZNr)**
> >
> > Reviewer EZNr raised this interesting point, and the author's response does not convince me. But I might be wrong.
> >
> > I assume that the authors will revise the final version of the paper to incorporate the points discussed in their response.

---

> ### Author Response · Authors · 2024-11-24
> **Thank you for your response + UPDATE with Bayes regression**
>
> Dear reviewer gpLZ,
>
> Thank you for getting back to us regarding our response to your review. We really appreciate your engagement in the rebuttal process.
>
> **UPDATE:** Following up on your comment regarding Bayesian regression, we have now included an additional experiment comparing (I)NPs with exact, model-based inference. These results are available in the Appendix A.8.4. of the updated manuscript. We hope that this addition will be of interest to the readers!
>
> In terms of the weather forecast experiment, naturally, we will make appropriate adjustments to the presentation of this experiment based on the final comments received.
>
> Once again, thank you for dedicating your time to reviewing our paper.
>
> Kind regards,
>
> The Authors

---

### Official Review · Reviewer_66xM · 2024-11-04

**Soundness:** 3
**Presentation:** 3
**Contribution:** 3
**Rating:** 8
**Confidence:** 3

**Summary:**

The work proposes a viewpoint on how to integrate prior knowledge into the learning process via informed meta-learning. The idea is to incorporate knowledge that is conditioned on the generative function, but is not a unique representation of the function. The meta-learning algorithm can then be applied to incorporate this knowledge. Neural Processes are used for empirical studies of the approach, demonstrating the advantages of prior knowledge in three specific cases.

**Strengths:**

The paper can potentially be applied to a lot of domains, where the expert knowledge and priors are presented.

The paper is overall clearly and well-written.

The proposed viewpoint is clearly illustrated based on a set of examples.

**Weaknesses:**

One of the core ideas of the paper’s perspective is to use prior knowledge K to guide the learning system. To assess how applicable this viewpoint is to different problems and domains, more discussion is needed on how to construct K for different scenarios, as the examples provided in the paper are very specific and not necessarily realistic. More discussion is also needed on how the properties of K influence the viewpoint. For example, in 5.1.1, more knowledge corresponds to a stronger prior. Can this be formalized for a general viewpoint?

Could you please comment on whether it is possible to have a stronger statement in Theorem 1 with a strict inequality?

Minor:

Line 245: "mete-training" -> "meta-training"

Please name the axes in Figure 6.

**Questions:**

For the MAML Example 1, how similarity between tasks is measured?

Could you please comment of how number of shorts or context points should be chosen while using the framework?

---

> ### Author Response · Authors · 2024-11-23
> **Response to your review -- part 1**
>
> We thank the reviewer for their insightful comments and constructive feedback. We have carefully considered each point of feedback and provide our point-by-point responses below.
>
> ---
>
> ### Weaknesses
>
>  **W1) Additional theory: strict inequality and more knowledge ⇒ stronger priors**
>
> Thank you for your questions regarding the theory which prompted us to include a deeper discussion on the conclusions from the presented Theorem 1. **UPDATE**: We have provided an extended theoretical discussion regarding the aspect of necessary conditions for the inequality of Theorem 1 to be strict, and a formalisation of the statement “more information leads to stronger priors”. This is now available in the Appendix A.1.3. of the updated manuscript. To summarise this discussion, we show that:
>
> 1. The inequality of Theorem 1 can be made strict under the assumption of positive mutual information between the target $y$ and knowledge $\mathcal{K}$ at the location $x$, formalised as $\mathbb{I}(y; \mathcal{K} | x) := \mathbb{H}[p(y \vert x)] - \mathbb{E}_{p(\mathcal{K})}\mathbb{H}[p(y \vert x, \mathcal{K})] > 0$
> 2. We can re-exprees the above in terms of the mutual information of $\mathcal{K}$ and $f$ to arrive at an equivalent condition of $\mathbb{I}(f; \mathcal{K}) - \mathbb{E}\_{p(y \vert x)}\mathbb{I}[(f; \mathcal{K} \vert x, y)] > 0$, where $\mathbb{I}(f; \mathcal{K}) = \mathbb{H}[p(f)] - \mathbb{E}\_{p(\mathcal{K})}\mathcal{H}[p(f \vert \mathcal{K})]$ and
> $\mathbb{I}(f; \mathcal{K} \vert x, y) = \mathbb{H}[p(f \vert x, y)] - \mathbb{E}\_{p(\mathcal{K})}\mathbb{H}[p(f \vert \mathcal{K}, x, y)]$.  Then, the condition $\mathbb{I}(y; \mathcal{K} | x) > 0 \Leftrightarrow \mathbb{I}(f; \mathcal{K}) - \mathbb{E}\_{p(y \vert x)}\mathbb{I}[(f; \mathcal{K} \vert x, y)] > 0$ requires that:
>     - The mutual information of $f$ and $\mathcal{K}$ is strictly positive, i.e. $\mathbb{I}(f ; \mathcal{K}) > 0$, which should trivially be satisfied for any valid knowledge representation by the defintion of knowledge as information about the ground-truth data generating process.
>     - Observing $y$ at a specified location $x$, brings sufficient information about the underlying $f$, e.g., the observational noise of $y$’s cannot be too high. If it is not the case, $\mathbb{E}_{p(y \vert x)}\left[\mathbb{I}(f; \mathcal{K} \vert x, y)\right]$ is high, bringing it closer to $\mathbb{I}(f; \mathcal{K})$ and thus making $\mathbb{I}(y; \mathcal{K})$ closer to 0.
> 3. We also formalise the meaning of more informative knowledge representations and how such more informative representation reduce the divergence between the posterior predictive $p(y \vert x, \mathcal{D}_C, \mathcal{K})$ and the ground-truth marginal $p(y \vert x, f)$.

---

> ### Author Response · Authors · 2024-11-23
> **Response to your review -- part 2**
>
> **W2) Construction and properties of knowledge representations**
>
> The construction of knowledge representations used for meta-training is an important aspect to be considered for a successful deployment of informed meta-learning in practice. This construction, is however, inherently application-specific and dependent on the representations of knowledge expected to be observed at test time—as provided by a domain expert or derived from a knowledge repository (wikipedia, textbooks). Further, the underlying meaning of knowledge representations that is learnable in our proposed data-driven manner is strictly dependent on the available data for meta-training. For instance, if we were to learn what “increasing” means in the context of regression tasks, our meta-training dataset must contain negative examples of non-increasing functions. To generalise this argument, we can assume that at test time we are interested in providing information that has up to $d$ intrinsic factors of variability, i.e. that the information contained in $\mathcal{K}$’s can be essentially compressed to a $d$-dimensional representation. In the synthetic regression example we have $d=3$ : slope, oscillation frequency, bias. In the example of bird classification, $d$  corresponds to the maximum number of attributes that a given knowledge representation can describe (eye color, beak length, etc.). To ensure that all properties of sample $f$’s associated with each of the $d$ underlying dimensions of $\mathcal{K}$ are learned adequatelly, the meta-training dataset should contain tasks that span this space sufficiently, as succesful extrapolation to previously unobserved regions of the knowledge space, let alone dimensions, cannot be guaranteed.
>
> The point about knowledge complexity and the resulting requirement of more training tasks for successful meta-learning has been mentioned in section 3.2.2 *Finite sample approximation* and demonstrated empirically in Appendix A.8.2. We also hint at the usage of generative models for synthetic data generation and augmentation as a plausible strategy for mitigating the challenge of meta-training set collection in section 3.2.2 *Existence of the meta-training set* and Appendix 2.2.2. This strategy has also been employed in our experiments on real-world datasets in sections 5.1 and 5.2, where knowledge representations are generated with GPT-4 aiming to simulate natural language descriptions that a real human expert may provide. We hope these experiments can inspire future applications across a wider range of domains, where natural language descriptions can guide numerical predictions such as healthcare, finance, or retail.
>
> **W3) Typos and axes in Figure 6**
>
> **UPDATE:** Thank you for pointing this out. We have now updated this in the manuscript.

---

> > ### Author Response · Authors · 2024-11-23
> > **Response to your review -- part 3**
> >
> > ### Questions
> >
> >  **Q1) Example 1) How similarity between tasks is measured?**
> >
> > We can think about the similarity between different learning tasks in terms of a distance metric between their optimal task-specific parameters. Let us denote by $\theta_i^*$ the set of weights minimising the generalisation error for a learning task $\tau_i$. Then similarity between $\tau_i$ and $\tau_j$ can be measured in terms of the distance $d(\theta_i^*, \theta_j^*)$ (e.g. the Euclidean distance is used to measure task similarity by Zhou et al.). The underlying assumption of conventional MAML, is that all tasks from our distribution are similar, so that by learning a common initialisation $\theta_0$, their task-specific parameters $\theta_i^*$ can be reached with just a few steps of gradient descent with respect to the task-specific observed data, $\mathcal{D}_i$. Knowledge-conditioned MAML, makes the learned initialisation a function of the task-specific knowledge, so that tasks with the same information are mapped to the same initialisations.
> >
> > Reference
> >
> > P. Zhou et al., Task similarity aware meta learning: theory-inspired improvement on MAML. *Proceedings of the Thirty-Seventh Conference on Uncertainty in Artificial Intelligence*, PMLR 161:23-33, 2021.
> >
> > **Q2) How number of shots / context points should be chosen while using the framework?**
> >
> >  Note, thanks to the usage of DeepSets, NPs can condition the learner on a varying number of context data points, therefore we need not to specify precisely the size of the context datasets a priori. The number of shots / context data points to be used for meta-training is dependent on the number of available datapoints in our tasks. Existing works in this field during meta-training sample context data points uniformly between 0 and $|\mathcal{D}^T|$ and we follow the same convention in our experiments (except for the temperatures experiment which is testing the ability of INPs for performing informed extrapolation). Naturally, at test time, the context data points should simply correspond to all of the available task-specific labelled data.
> >
> > ---
> >
> > We greatly appreciate the time and effort you’ve dedicated to reviewing our paper. We hope that our proposed changes and responses address your concerns satisfactorily. Please don’t hesitate to let us know if any further clarifications are required.
> >
> > Thank you for helping us improve the quality of this work.
> >
> > The authors of submission #11605

---

> > > ### Comment · Reviewer_66xM · 2024-11-26
> > > **Response to comments**
> > >
> > > I would like to thank the authors for their response and updates to the manuscript. The new theoretical analysis helps to provide a clearer understanding of when the knowledge is strictly helpful, and the conclusions and discussions in the new Appendix section are reasonable.
> > >
> > > Regarding the knowledge, while keeping it ambiguous helps to define the general concept, a more detailed and nuanced discussion could inspire more specific future research. At the very least, I encourage the authors to include the table compiled in response to Reviewer gpLz's comment in the manuscript.
> > >
> > > Additionally, please note that there is a broken Figure link on line 1783.
> > >
> > > I have updated my scores accordingly.

---

> > > > ### Author Response · Authors · 2024-11-27
> > > > **Thank you!**
> > > >
> > > > Dear reviewer 66xM,
> > > >
> > > > Thank you for updating your score, we really appreciate your acknowledgment of our work.
> > > >
> > > > Based on your comment, we have now included the table with example applications in the updated manuscript and fixed the reference to Figure A.5--thank you for spotting this!
> > > >
> > > > Once again, thank you for your engagement in the rebuttal process and your help in improving the quality of our submission.
> > > >
> > > > Kind regards,
> > > >
> > > > The authors

---

### Official Review · Reviewer_EZNr · 2024-11-08

**Soundness:** 3
**Presentation:** 3
**Contribution:** 2
**Rating:** 6
**Confidence:** 4

**Summary:**

The paper provides a theoretical framework around conditional meta-learning, oriented toward conditioning neural processes on natural language info about temporal processes.

**Strengths:**

Nice concrete examples through the theory; good explanations and clear notation.

**Weaknesses:**

Overall an interesting and well-written paper, but  seems like it's maybe missing references to conditional metalearning, and doesn't compare to the baselines it discusses early on. The theory seems like a lot of math to say things that are kind of vacuous, but such is math, and it's well explained with nice concrete examples. The biggest criticism I have is that the way K is given to the model, concretely, is very unclear, and in my experience (conditioning physics-informed neural nets, which come to think of it is another area that should be cited as relevant lit) this kind of 'technical detail' makes a huge practical difference.

**Questions:**

Detailed comments by section:

- I would add "particularly" to the first sentence of the abs
 - almost very nice first fig, but caption could be improved (what are the different indices on Ks and how do they relate to K

2.
- nice concrete examples of K in the problem setting
- not sure exactly how to phrase this, but it seems worth mentioning how strong the assumption of an underlying stochastic process is. e.g. in the nice explanatory figure, in the observable data space you have a sine and a line, and these are clearly different, but what we actually have in real life is often a bunch of high-D points that you could fit a sine or a line to with equal error and it doesn't mean that either of them is properly the "underlying" function. I'm not trying to say this assumption isn't fine to make, but I think it's worth emphasizing how abstracted it is from "real" data
  - similarly for the assumption of conditional independence of K and D that is necessary for the proof; it seems to me like it's a lot of math to say something pretty obvious; that if we add new necessarily true and relevant information that will improve things ... it would be nice to give an idea of how much the proof relies on those properties.

3.
 - nice concrete examples but man the formatting is hard to read; please make it not all in italics!
 - the remarks section is a bit random and take up a lot of room relative to what they provide; suggest moving to appendix to make more room for easier to read formatting and extra concrete examples/explanations of experimental results and connections from theory to experiments.

4
- the arrangement of the sections makes sense, but it does mean all the theory and acronyms about neural processes are really far from the empirical results; it would be nice to tie them together better with cross-references
- unclear how the aggregator operator works

5
- definitely nice to have the empriical results but because of the synthetic setting it's hard to take anything from the results without the discussion being more concrete -- it would be nice to continue the concrete example pattern you had from the theory section (e.g. for the dist shift experiments, what was the K you provided?).
 - why not use real weather forecasts?? introducing a LLM adds an unnecessary layer of complexity and potential for hallucinations
- again unclear how the K is actually informing the NP -- are you encoding it with a pretrained LLM and appending that to the input of the neural process? adding? something else? these details are the technical core of the paper and it's important to make them clear.

---

> ### Author Response · Authors · 2024-11-23
> **Response to your review -- part 1**
>
> Thank you for taking the time to review our work. We appreciate the reviewer’s detailed feedback on our submission and we address their concerns point by point in our response below.
>
> ---
>
>
> ## Weaknesses
>
>  **W1) Missing references to conditional meta-learning**
>
> References to conditional meta-learning, while not explicitly named as such, have been made in the Appendix A.4. Extended Related Work under the M*eta-learning with meta-data* subsection (e.g.: Kaddour et al., Tseng et al., Iwata & Kumagai). We agree, however, that the main body of the paper should include an explicit reference as well.
>
>  **UPDATE:** Following the reviewer’s comment we have now included additional references to conditional meta-learning, including them in the main body of the paper when conditional meta-learning is mentioned for the first time and in the related work section.
>
> **W2) Lack of comparisons to ‘baselines’ discussed earlier on**
>
>  **Automated vs. model-based approaches.** The alternative methods for knowledge integration discussed in the paper are model based and necessitate the involvement of a human engineer to interpret the knowledge in a given representation $\mathcal{K}$ and manually encode it within the learning algorithm (lines 112-113). This dependence on manual algorithm design is particularly limiting in cases where the underlying functional form of the data generating processes is unknown, making conventional knowledge integration challenging and n many cases simply infeasible. In contrast, the focus of this paper is on automating the knowledge integration process. This  is especially relevant for applications where such functional forms are complex or undefined, and the task-specific knowledge is difficult to represent in a form of an exact mathematical equation or a constraint—e.g. qualitative proporties described in human natural language. To the best of our knowledge, the framework of informed meta-learning presented here is the only viable approach for achieving fully automated and controllable integration of such forms knowledge.
>
>  **INPs vs. alternative implementations of informed meta-learning.** Within this paradigm, we present the method of Informed Neural Processes method primarily for illustrative purposes, aiming to inspire further research in this area. Whilea systematic evaluation of alternative knowledge-conditioned meta-learning methods, including conditional MAML (Example 1) and approaches like in-context learning with LLMs (lines 527-528), may be valuable for practitioners, such a benchmarking effort would constitute a substantial undertaking that could warrant its own dedicated study. The current paper thus aims to serve as a foundational reference point for future work.
>
> **W3) The way K is given to the model is unclear**
>
> We made a conscious decision of keeping this paper conceptual in nature—with its main contributions being the proposed new way of thinking about meta-learning as an enabler for automated and controllable inductive bias specification and the accompanying theory. The implementation details have been purposefully postponed to the Appendix, where for every experiment, we provide details on how knowledge is encoded (see Appendix A.5 and A.7).
>
>  **UPDATE:** To strike a better balance between theory and practice we have included more detail about the implementation details in the main body of the paper. Following the reviewer’s suggestion we have included a short explanation on the aggregating operator when it is introduced in the main body of the paper—in most experiments the aggregation is a simple summation of the data embedding and the knowledge embedding vectors.
>
>
>  **W4) Conditioning physics-informed neural nets should be cited**
>
>  Thank you for your comment regarding conditioning physics-informed neural networks. In fact, the paper already includes references to conditional meta-learning for PINNS in the Appendix A.4. Extended Related Work under the M*eta-learning with meta-data* subsection (Huang et al. (2022); Belbute-Peres et al. (2021)). **UPDATE:** We have included an additional reference to conditional PINNs in our related work section.

---

> ### Author Response · Authors · 2024-11-23
> **Response to your review -- part 2**
>
> ## Questions
>
> **Q1) *I would add "particularly" to the first sentence of the abs***
>
> **UPDATE** Thank you for your suggestion, this has now been added in the abstract.
>
>  **Q2) Fig. 1 caption (indicies on Ks)**
>
> **UPDATE** Thank you for your suggestion, we have now added a brief comment explaining the relationships between the function and knowledge indices.
>
> **Q3) Assumptions of the stochastic process and conditional independence.**
>
> **The stochastic process convention.** The assumption that an underlying function is sampled from a stochastic process is powerful and very common in probabilistic models, especially in nonparametric Bayesian approaches like Gaussian Processes (GPs), allowing models to naturally handle uncertainty in their predictions. While modelling with GPs can be restrictive, due to the strong assumption of the stochastic process being a Gaussian, methods like Deep Kernel Learning or Neural Process, enable far greater flexibility. In particular, NPs discussed in this paper, can approximate arbitrarily complex stochastic processes that are learned based on real-world data, imposing little to no assumptions on the properties of sample functions.
>
> **Model uncertainty.** To address your example with the sinusoid vs. linear function, the fact that many different sample functions can fit the observed data equally well is exactly the problem of inductive bias specification in low sample regimes. With only a few observations we are unable to decide with certainty whether a linear or a sinusoid function is a better fit, unless we posses additional expert knowledge about the underlying data generating process, or we can collect more data. This, however, does not imply that there does not exist an underlying function $f$ that gave rise to these observations. The assumption we make here and which is the underlying principle of most of machine learning approaches, is that $f$ can be recovered with certainty as the number of observed data points grows.
>
>  **Conditional independence.** In terms of the conditional independence assumption, we argue that this is a rather a natural assumption to make. Namely, if we know the ground-truth form of $f$ that gave rise to the noisly-observed dataset $\mathcal{D}$, in order to describe the properties of $f$, we need not observe individual datapoints $(x_i, y_i) \in \mathcal{D}$. The information contained in $\mathcal{K}$, can be derived purely from $f$.
>
>  **UPDATE** Thank you for raising these questions. We appreciate that the assumptions made in our paper may seem restrictive or only useful for the theory. We have now added additional references and intuitions behind these assumptions to motivate their validity.
>
> **Q4) Nice concrete examples in section 3. but formatting is hard to read**
>
>  **UPDATE** Thank you for your comment. We have used the default theorem latex environment to define the example environment. We agree, however, that the italics make it difficult to read and we have changed it to a standard font in the update version of the manuscript.
>
> **Q5) INP and NP acronyms.**
>
>  **UPDATE** Thank you for this suggestion. We have now made sure that the NP and INP acronyms are properly introduced in section 4 so that they are better linked across sections.
>
> **Q6) Unclear how the aggregator operator works.**
>
> Addressed in **W3)**.
>
>  **Q7) For the dist shift experiments, what was the K you provided?**
>
> As specified around the lines 356-357, the knowledge presented in the distribution shift experiment specifies the ground-truth value of the parameter $b$

---

> > ### Author Response · Authors · 2024-11-23
> > **Response to your review -- part 3**
> >
> > **Q8) Why not use real weather forecasts?**
> >
> >  We would like to point out that real-world weather forecasts cannot be seen as representations of knowledge—these are made before observing the true values of the temperatures on the given day, and therefore are likely to come with a significant error in comparison to the ground-truth temperature values (e.g. the forecast may say that the temperatures will rise, but the data points may indicate the opposite). To learn a faithful mapping between the knowledge space and the function space that aligns with our human understanding, we require that information contained in the individual representation of knowledge for each task is in agreement with the underlying data generating process. Therefore, during meta-training, we use knowledge representations that are derived from the ground-truth observations, ensuring that alignment between observed data and knowledge representations holds. After meta-training is completed, at test time, human-written knowledge representations may be used.  In this particular experiment, we have manually examined a random sample of descriptions generated by the LLM and compared them with the data to ensure that the LLM’s output is indeed correct with respect to the observed values of the temperature.
> >
> > Obtaining a large dataset of knowledge representations for each task in the meta-training set from human experts can be costly and labour intensive. Going forwards, in real-world deployment of informed meta-learning we expect LLM-based synthetic data generation or augmentation to play a significant role in the meta-training pipeline.  As mentioned in Appendix A.3.2, we propose that LLMs are used to generate synthetic representations of knowledge to be used during meta-training, simulating knowledge representations expected to be provided by human experts at test time.
> >
> > **Q9) How K is informing the NP?**
> >
> > Addressed in **W3)**.
> >
> > ---
> >
> > Once again, thank you for your thoughtful comments and questions. We hope that our proposed changes and responses address your concerns satisfactorily. Please don’t hesitate to let us know if any further clarifications are required.
> >
> > Thank you for helping us improve the quality of this work.
> >
> > The authors of submission #11605

---

> > > ### Author Response · Authors · 2024-11-29
> > > **Thank you for your feedback**
> > >
> > > Dear Reviewer EZNr,
> > >
> > > Thank you for your valuable and detailed feedback on our work. We’ve carefully considered your questions and concerns and made several updates to our submission. Given the limited time remaining in the discussion period, we would greatly appreciate it if you could let us know whether our responses and updates sufficiently addressed your concerns. If there’s any additional clarification or information we can provide to assist in your evaluation, please don’t hesitate to let us know!
> > >
> > > Thank you again for your time and effort in the review process—it is truly appreciated.
> > >
> > > Kind regards,
> > >
> > > The Authors

---

### Author Response · Authors · 2024-11-23
**Global response**

We would like to express our gratitude to all the reviewers for their constructive feedback and insights on our submission.

We are encouraged by the reviewers' recognition of our work. The tackled research question was described as “interesting, as it connects a classic topic in machine learning research—how to integrate human domain knowledge into the algorithm's inductive bias—with recent developments (LLMs)" (**gpLZ**). The proposed approach was noted for its broad potential applicability, as it "can potentially be applied to a lot of domains, where the expert knowledge and priors are present." (**66xM**). The paper was also recognised for making "interesting suggestions for how novel developments in LLMs can inform existing work in knowledge integration / meta-learning." **(gpLZ**).

The reviewers also found the paper to be "overall clearly and well-written"(**66xM**), with "good explanations and clear notation." (**EZNr**) and highlighted the inclusion of "nice concrete examples through the theory" (**EZNr**) and that the "proposed viewpoint is clearly illustrated based on a set of examples" (**66xM**) and “experiments are well-designed” (**gpLZ**).

We thank the reviewers for their positive feedback and appreciate the opportunity to clarify and enhance our manuscript based on the comments received. Below we outline the key actions taken in response to the reviewers’ comments. Remaining queries have been addressed in individual responses under each review.

### Summary of key actions taken

In response to reviewer 66xM's questions, we have extended the theoretical discussion following Theorem 1. This now includes an analysis of the conditions under which the presented inequality becomes strict, as well as a formalisation of the intuition that more informative knowledge representations lead to improved predictions. These additions can be found in Appendix 1.3.3 of the updated manuscript. Additionally, we have addressed minor typos, made stylistic changes, and updated the references.

### Updated manuscript

Based on the helpful and detailed feedback of the reviewers we have updated our manuscript and uploaded the revised version. All changes made are highlighted with a blue text color. We hope that the proposed updates improve the paper’s clarity, ease of understanding, and that they address the reviewers’ questions and concerns.

---

We are grateful for the reviewers' feedback, which helped us improve the presentation of our work. We are open to further discussions to clarify any aspects of our submission.

Kind regards,

The authors of submission #11605

---

### Meta-Review · Area_Chair_NBRq · 2024-12-23

**Metareview:**

This paper explores how prior knowledge represented (e.g. in natural language) can be incorporated into machine learning models. Key idea centers around informed meta-learning (i.e. learning to learn).  This is a theoretically leaning paper and describes the principles of informed meta-learning and inductive bias selection.

I recommend acceptance as all reviewers are positive about the approach - the biggest criticisms centers around clarity and empirical evaluation. The idea itself is novel and sufficient discussion is provided to analyze the approach.

**Additional Comments On Reviewer Discussion:**

The authors provided a rich rebuttal and one round of discussion with two reviewers materialized. The authors addressed many of the questions on clarity and methodology and convinced the two reviewers on the merit.

---

### Decision · Program_Chairs · 2025-01-22

Accept (Spotlight)